## Scientific Report

# Dual role of USP30 in controlling basal pexophagy and mitophagy

Elena Marcassa[1], Andreas Kallinos[1], Jane Jardine[1], Emma V Rusilowicz-Jones[1], Aitor Martinez[1], Sandra Kuehl[2], Markus Islinger[2], Michael J Clague[1],* & Sylvie Urbé[1],**

## Abstract

USP30 is an integral protein of the outer mitochondrial membrane that counteracts PINK1 and Parkin-dependent mitophagy following acute mitochondrial depolarisation. Here, we use two distinct mitophagy reporter systems to reveal tonic suppression by USP30, of a PINK1-dependent component of basal mitophagy in cells lacking detectable Parkin. We propose that USP30 acts upstream of PINK1 through modulation of PINK1-substrate availability and thereby determines the potential for mitophagy initiation. We further show that a fraction of endogenous USP30 is independently targeted to peroxisomes where it regulates basal pexophagy in a PINK1- and Parkin-independent manner. Thus, we reveal a critical role of USP30 in the clearance of the two major sources of ROS in mammalian cells and in the regulation of both a PINK1-dependent and a PINK1-independent selective autophagy pathway.

**Keywords** mitochondria; peroxisomes; PINK1; ubiquitin; mitophagy; USP30
**Subject Categories** Autophagy & Cell Death; Post-translational Modifications, Proteolysis & Proteomics

## Introduction

Loss of function mutations in the E3 ligase Parkin and the kinase PINK1 are the most common known causes of familial early onset Parkinson's disease (PD) [1]. Both proteins play a key role in the controlled clearance of damaged mitochondria by mitophagy [1,2]. This process is thought to involve the engulfment of dysfunctional mitochondrial fragments in an autophagic membrane for safe disposal in lysosomes. Failure to clear damaged mitochondria results in the accumulation of toxic reactive oxygen species (ROS) and cell death, in particular of the dopaminergic neurons of the *substantia nigra* in PD patients. From a mechanistic point of view, the key signal initiating mitophagy is the generation of phospho-ubiquitin by PINK1 [3]. PINK1 is usually unstable but accumulates at depolarised or otherwise compromised mitochondria [4]. Phospho-ubiquitin chains can directly recruit specific autophagy receptors, optineurin and NDP52, which in turn engage the autophagy machinery and nascent autophagic membranes [4,5]. Phospho-ubiquitin also activates Parkin, which itself becomes phosphorylated by PINK1 on a topologically equivalent residue in its UBL domain. This leads to further ubiquitylation of outer mitochondrial membrane (OMM) proteins and amplification of the autophagy signal [4,6–9].

USP30 is the only deubiquitylase (DUB) constitutively associated with the OMM, where it has been shown to counteract Parkin-dependent mitophagy by deubiquitylating OMM proteins, in particular TOMM20 [10–14]. Depletion of USP30 in Parkin-overexpressing cells promotes the clearance of mitochondria in response to mitochondrial depolarising agents [10,12]. This suggests that USP30 may be an attractive target for therapeutic approaches in PD. However, much of the current data rely on cells that are engineered to overexpress large amounts of Parkin and which are subjected to an acute depolarising trigger. Much less is known about the relevance of USP30 function in unperturbed cells expressing limiting amounts of endogenous Parkin, which precludes the whole-scale clearance of the mitochondrial network and thus cannot be monitored by straightforward Western blotting techniques.

Here, we have made use of two previously described fluorescent mitophagy reporter systems to monitor basal (constitutive) mitophagy in the absence of Parkin overexpression. We show that USP30 regulates basal mitophagy in a PINK1 (but not Parkin-)-dependent manner, whilst PINK1 depletion on its own has no effect. Our data lead us to propose a new model that places USP30 upstream of PINK1, where it could determine the threshold for mitophagy initiation by tonically suppressing basal ubiquitylation of specific outer mitochondrial membrane proteins. This model reconciles the reported poor activity of USP30 on phosphorylated ubiquitin chains [15,16] with its ability to modulate PINK1-dependent mitophagy [10,12]. Intriguingly, we find a separate pool of USP30 associated with peroxisomes where it limits the basal level of pexophagy in a manner that does not require PINK1 function. Thus, we reveal a critical role of USP30 in the clearance of the two major sources of ROS in mammalian cells and in the regulation of both a PINK1-dependent and a PINK1-independent selective autophagy pathway.

---

1  Cellular and Molecular Physiology, Institute of Translational Medicine, University of Liverpool, Liverpool, UK
2  Institute of Neuroanatomy, Centre for Biomedicine and Medical Technology Mannheim, Medical Faculty Mannheim, University of Heidelberg, Mannheim, Germany
   *Corresponding author. Tel: +44 151 7945308; E-mail: clague@liv.ac.uk
   **Corresponding author. Tel: +44 151 7945432; E-mail: urbe@liv.ac.uk

# Results and Discussion

### Enhancement of basal mitophagy by USP30 depletion is dependent on PINK1

We previously showed that depletion of USP30 in Parkin-overexpressing RPE1 cells accelerates the depolarisation-induced ubiquitylation and degradation of TOMM20 in a PINK1-dependent fashion [12]. The expression level of Parkin in these cells by far exceeds the endogenous levels seen across panels of cell lines. Moreover, in order to observe a synchronised and complete clearance of mitochondria, an acute depolarising trigger, for example CCCP or a combination of antimycin A and oligomycin A, is required. Naturally occurring, sporadic mitophagy events in unperturbed cells can be monitored by mitochondrially targeted, pH-sensitive fluorescent reporters that respond to the acidic environment of lysosomes, the final destination of eliminated mitochondrial remnants. Two main systems have been reported: a mitochondrial matrix-targeted Keima reporter (mt-Keima), which changes its excitation profile in response to pH [17], and an OMM-targeted mCherry-GFP-Fis1(101-152) chimera (MGFIS) [18]. The latter responds to low pH with a loss of green fluorescence leading to the emergence of "red only" lysosomal punctate structures comprising mitochondrial remnants (mitolysosomes), which provide a quantitative measure of mitophagy events.

Depletion of USP30 in U2OS-MGFIS cells using two individual siRNAs (D1 and D3) results in a clear increase in mitophagy over a 72-h period (Fig 1A and B). Separating cells into categories according to the number of "red" punctate structures (dots) allows us to capture more accurately the response of individual cells within the population. This reveals a highly significant shift in the number of red puncta per cell (Fig 1B). Concomitant depletion of ATG7 abolished the enhancement of mitophagy seen in USP30-depleted cells, indicating that this process is contingent on a canonical core component of the autophagy machinery (Fig 1C).

We wondered whether, by analogy with the role of USP30 in Parkin-overexpressing cells, this increase in mitophagy was dependent on PINK1. Depleting PINK1 did not affect basal levels of mitophagy, in agreement with two recent reports in mitophagy reporter flies and mice [19,20]. However, PINK1 depletion did suppress the enhancement of mitophagy seen upon USP30 knockdown (Fig 1D).

We did not observe a corresponding requirement for Parkin, which is already undetectable by Western blotting in these cells (Fig EV1A). Given that FIS1 has been shown to localise to both mitochondria and peroxisomes, a small fraction of the MGFIS1 mitophagy reporter could in principle be targeted to peroxisomes [21]. Thus, we further validated these experiments in RPE1 cells that were transiently transfected with the mt-Keima reporter, which is targeted to the mitochondrial matrix. USP30 depletion again enhanced basal mitophagy approximately twofold, and this effect is contingent on PINK1 (Fig 1E). Note that RPE1 cells, like U2OS cells, do not express detectable levels of Parkin. Taken together, these results suggest that USP30 depletion unmasks a distinct component of basal mitophagy that is PINK1- but not Parkin-dependent and is tonically suppressed by USP30 in control cells.

Our data showing that USP30 suppresses a PINK1-dependent component of basal mitophagy indicate a link between USP30 function and phospho-ubiquitin, the key substrate of PINK1 in mitophagy. However, USP30 itself is unable to process phosphorylated ubiquitin chains efficiently [15,16]. Our epistatic findings are compatible with a working model in which USP30 does not merely act downstream of PINK1 and Parkin, but rather restricts the basal ubiquitylation status of one or several putative trigger OMM proteins in range of PINK1. This limits PINK1 substrate and in turn the generation of recruitment sites for selective autophagy receptors (and Parkin). PINK1 has a high turnover rate, which is contingent on its engagement with the mitochondrial import system and as a result it continuously surveys the mitochondrial environment. Constitutive USP30 activity may thus serve to prevent unscheduled initiation of mitophagy, whilst allowing the system to be held at a hair trigger. In this scenario, USP30 acts upstream of PINK1 to set the threshold for induction of the mitophagy process and its depletion leads to an increase in the incidence of mitophagy events (Fig 1F).

### USP30 localises to peroxisomes

Imaging USP30-GFP at low expression levels in either U2OS or hTERT-RPE1 cells, we noticed a separate population of punctate structures in addition to the mitochondrial localisation that has previously been observed [13,14]. These punctate structures were reminiscent of peroxisomes, single membrane organelles that share

---

**Figure 1.  Enhancement of basal mitophagy by USP30 depletion is dependent on PINK1.**

A  Representative images of U2OS cells stably expressing mCherry-GFP-Fis1$_{(101-152)}$ (U2OS-MGFIS), transfected with non-targeting (NT1) or USP30 targeting siRNA (D1, D3) for 72 h prior to imaging. Scale bar 10 μm.

B  Quantification of mitolysosomes (red puncta, "dots") in U2OS-MGFIS cells, treated as in (A). Average ± SD; $n$ = 6 independent experiments, 40 cells/experiment; left: one-way ANOVA and Dunnett's multiple comparison's test, ***$P < 0.001$; right: two-way ANOVA and Bonferroni's multiple comparison test, *$P < 0.05$, **$P < 0.01$, ***$P < 0.001$.

C  Quantification of mitolysosomes in U2OS-MGFIS cells transfected with siRNAs targeting either USP30 (D1) or ATG7 or both. Average ± SD, $n$ = 3 independent experiments, 40 cells per experiment; one-way ANOVA and Dunnett's multiple comparison's test, *$P < 0.05$. Also shown is a representative Western blot.

D  U2OS-MGFIS cells were treated with siRNA and analysed as in (B). Average ± SD; $n$ = 3 independent experiments; 40 cells per experiment; one-way ANOVA and Dunnett's multiple comparison's test, *$P < 0.05$. Also shown is a representative Western blot.

E  Quantification of mitolysosomes in hTERT-RPE1 cells transfected with non-targeting (NT1) or USP30 (D1, D3) or PINK1 targeting siRNA and mt-Keima. Average ± SD; $n$ = 3 independent experiments; 40 cells per experiment; one-way ANOVA and Dunnett's multiple comparison's test, *$P < 0.05$, **$P < 0.01$. Also shown is a representative Western blot.

F  Proposed model: USP30 acts upstream of PINK1 and limits the basal ubiquitylation (U) of outer mitochondrial membrane proteins, which serve as a substrate for PINK1. Phospho (P)-ubiquitin binds to specialized autophagy adapters (optineurin and NDP52) leading to activation and recruitment of the autophagy machinery including LC3-decorated autophagic membranes.

Source data are available online for this figure.

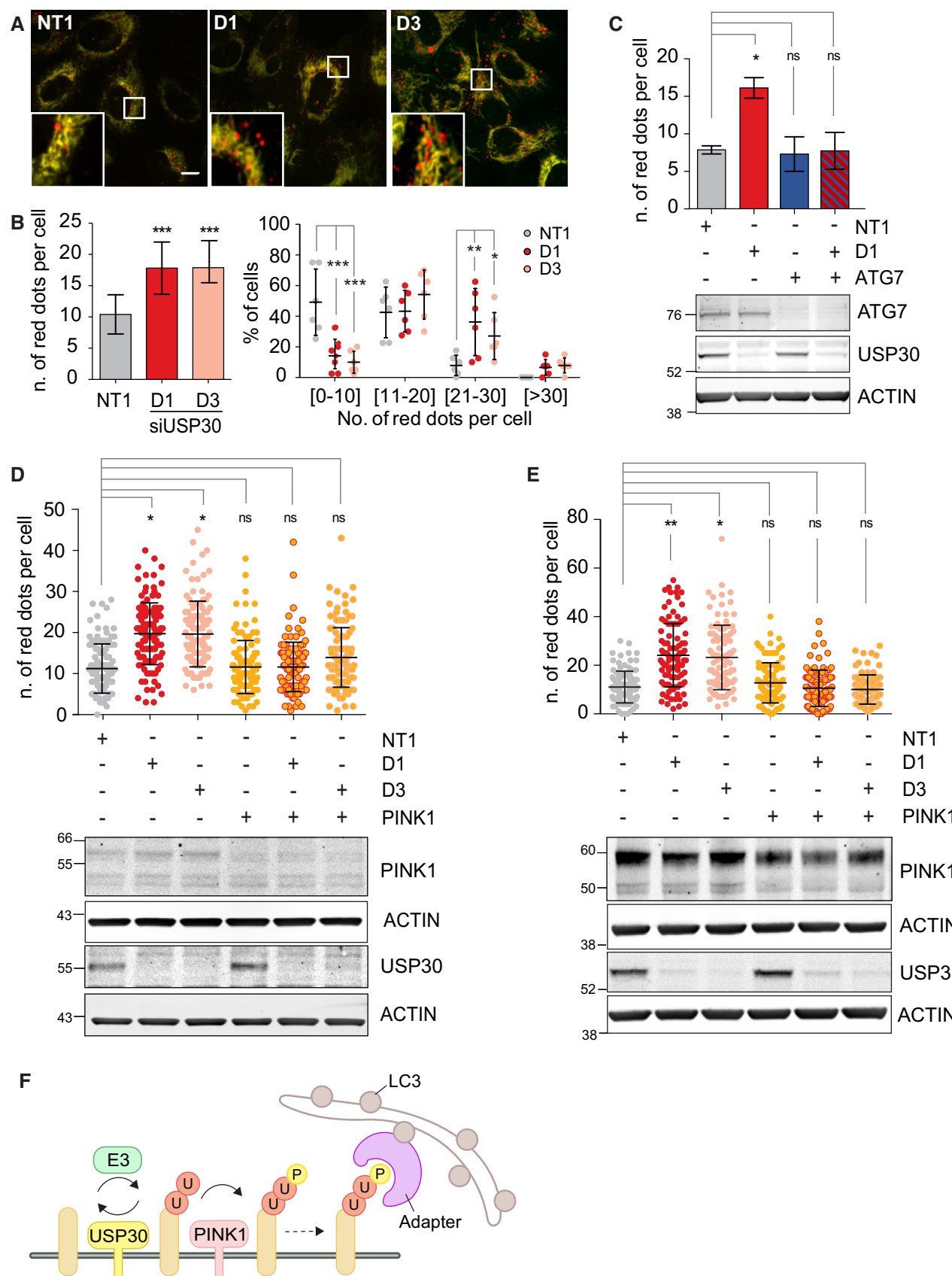

**Figure 1.**

with mitochondria a partially overlapping proteome and interlinked metabolic functions [22–24]. The punctate pool of USP30 co-localises well with peroxisomal proteins, catalase and PMP70 (Figs 2A and EV2A and B). Peroxisomal matrix proteins encode a C-terminal consensus motif (SKL) that can be used to target fluorophores to the matrix of peroxisomes. Live-cell imaging of Cherry-SKL marked peroxisomes in USP30-GFP expressing cells provides further confirmation for peroxisomal targeting of USP30 and illustrates the dynamic interactions between the mitochondrial network and peroxisomes (Fig 2A and Movie EV1). Immunofluorescence microscopy of SHSY5Y cells allowed us to visualise endogenous USP30, which is more abundant in these cells. In addition to the well-characterised mitochondrial localisation, we detected a siRNA-sensitive pool of USP30 that is associated with peroxisomes (Fig 2B and C). Further confirmation of the association of an endogenous pool of USP30 with peroxisomes was obtained using a well-established, orthogonal subcellular fractionation approach (Fig 2D and E; [25,26]). Density gradient centrifugation facilitates the separation of peroxisome-enriched high density fractions (Fig 2D, L1, L2) from the lighter mitochondrial fractions (Fig 2D, L5, L6). Given that the surface area of mitochondria, even in HepG2 cells, by far exceeds that of peroxisomes, it is not surprising that the bulk of USP30 co-migrates with mitochondria. However, we also detect a distinct peak of USP30 in the peroxisomal fraction (Fig 2D). Overall, the distribution of USP30 across the gradient is very comparable to that of glutathione-S-transferase κ (GSTK1), which is also shared by peroxisomes and mitochondria [27].

## Targeting of USP30 to peroxisomes is independent of mitochondria

We wondered how USP30 reaches the peroxisomes and first considered the possibility of prior insertion into mitochondria followed by shuttling to peroxisomes. McBride and colleagues have described a pathway linking mitochondria and peroxisomes that relies on the formation of mitochondrial-derived vesicles and requires the core component of the retromer complex, VPS35 [28]. This pathway has previously been proposed to mediate the transfer of the ubiquitin E3 ligase Mul1 (also called Mulan or MAPL) from mitochondria to peroxisomes [28,29]. USP30-GFP, transfected into cells that had first been efficiently depleted of VPS35, still localised to peroxisomes (Fig 3A). More recently, a mitochondrial-derived vesicle pathway that does not involve VPS35 has also been implicated in the *de novo*

formation of peroxisomes [30]. We thus conceived of an experiment that would allow us to fully discount the involvement of mitochondria. For this purpose, we made use of YFP-Parkin-overexpressing hTERT-RPE1 cells that we and others have previously used to study depolarisation-induced mitophagy [12,31]. Treatment of these cells with oligomycin A and antimycin A (OA) results in a complete elimination of mitochondria within 24 h without affecting peroxisome abundance or distribution (Fig 3B). USP30-RFP, subsequently introduced into these cells, is still able to reach peroxisomes. Taken together, our data suggest that a pool of USP30 can be targeted to peroxisomes independently of mitochondria. The same approach also allowed us to address the question whether the endogenous USP30 is targeted to the membrane or the matrix of peroxisomes. Upon OA-induced depletion of mitochondria from RPE1-YFP-Parkin cells, the residual predominantly peroxisome associated pool of USP30 displays integral membrane protein properties (c.f. PMP70), mirroring its behaviour in untreated cells, where the bulk is associated with mitochondria (Figs 3C and EV2C and D). Protease protection assays further indicate that its catalytic domain is exposed to the cytosol (Fig 3D).

## Different sequence requirements for mitochondrial and peroxisomal targeting of USP30

What then is the targeting determinant that dictates insertion into the peroxisomal membrane? Catalytic activity is dispensable for peroxisomal targeting as a catalytically inactive USP30 mutant (C77S) shows a similar degree of colocalisation with peroxisomes (and mitochondria) to the wild-type protein (Fig 4A). Although USP30 is not a tail-anchored (TA) protein, its short transmembrane domain, framed by conserved stretches of basic residues, is reminiscent of other well-characterised dually targeted mitochondrial and peroxisomal TA proteins that engage the shuttling receptor PEX19 [32]. We next generated a series of truncation mutants and established that the N-terminal region encompassing amino acids 1–53 including the transmembrane domain is both necessary and sufficient for targeting USP30 to peroxisomes (Fig 4B and C). Insertion into the OMM requires the addition of a short cytoplasmically orientated polybasic stretch (present in USP30(1–68)) that has previously been shown to be critical for mitochondrial localisation within the context of the full-length protein (Fig 4D) [13]. Thus USP30 localisation to mitochondria and peroxisomes relies on distinct targeting sequences.

---

**Figure 2. USP30 localises to peroxisomes.**

A    hTERT-RPE1 cells were transfected with USP30-GFP for 24 h, then fixed and stained for catalase and PMP70, or cotransfected with mCherry-SKL and fixed. Scale bars 10 μm.

B    Representative images of SHSY5Y cells transfected for 144 h with non-targeting (NT1) or USP30 siRNA (D3), fixed and immunostained for endogenous USP30 (AlexaFluor488; green) and PMP70 (AlexaFluor594; red). Scale bar 10 μm. I and II; enlarged insets with 2.5-μm scale bars. Arrows highlight individual PMP70-peroxisomes that stain positive for USP30.

C    Representative Western blot of cells shown in (B).

D, E    HepG2 subcellular fractionation and density gradient separation of "light mitochondrial" fractions (LM). USP30 is present in both peroxisomal (L1, L2) and mitochondrial (L5, L6) fractions. HM fractions from wild-type (WT) and USP30 KO RPE1 cells were loaded alongside the HepG2 fraction for easy identification of the USP30 band. The asterisk marks a non-specific band retained in USP30 KO cells. ACOX1-A, acyl CoA oxidase 1 A-band; GSTK1, glutathione-S-transferase κ-1; VDAC1, voltage-dependent anion-selective channel protein 1; Po, peroxisome marker, Mito, mitochondria marker; PNS, post-nuclear supernatant; HM, heavy mitochondria fraction; Mic, microsome fraction; Cyt, cytosolic fraction; L1-L6, LM gradient fractions.

Source data are available online for this figure.

                    

**Figure 2.**

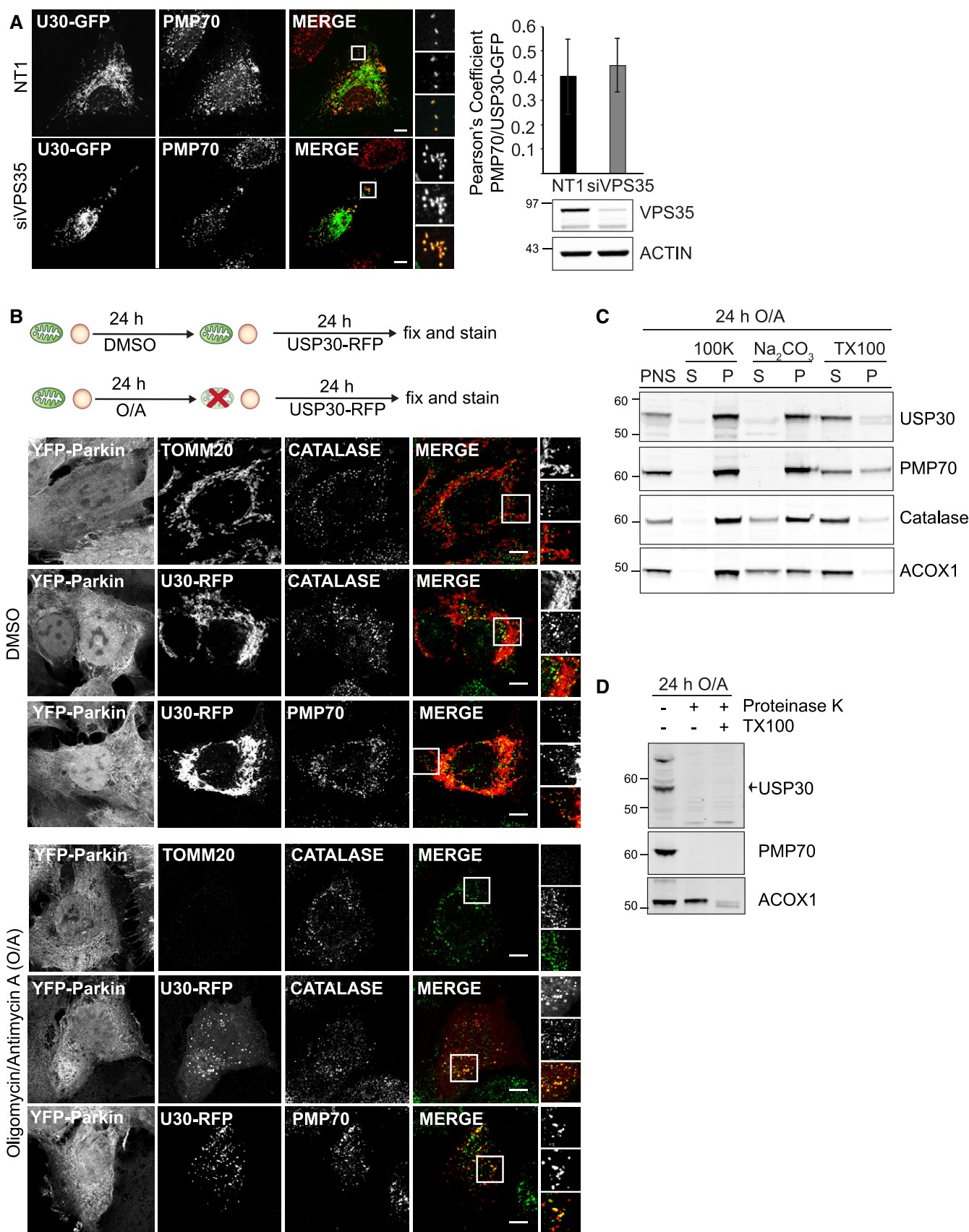

**Figure 3.**

◀

**Figure 3.  USP30-GFP is targeted to peroxisomes independently of mitochondria.**

A   hTERT-RPE1 cells were transfected with non-targeting (NT1) or VPS35 siRNA for 120 h, then transfected with USP30-GFP for 24 h and subsequently stained for PMP70. Shown are representative images and Pearson's correlation coefficients (full z-stack, mean ± SD, n = 10 cells). Scale bars 10 μm.

B   hTERT-RPE1 YFP-Parkin cells were first treated for 24 h with oligomycin A (1 μM) and antimycin A (1 μM) or DMSO, then transfected with USP30-RFP and fixed 24 h post-transfection. Cells were stained for catalase or PMP70 (AlexaFluor350, green) alone. A set of untransfected cells treated in parallel were co-stained for TOMM20 (AlexaFluor555, red). Scale bars: 10 μm.

C   Peroxisomal USP30 is an integral membrane protein. Membrane fractions derived from hTERT-RPE1 YFP-Parkin cells, treated for 24 h with oligomycin A (1 μM) and antimycin A (1 μM), were extracted either with 0.1 M alkaline $Na_2CO_3$ or 2% Triton X-100 and 1 M NaCl, and supernatants and pellets were analysed by SDS–PAGE and immunoblot. Representative experiment (n = 2).

D   Topology of peroxisomal USP30. A post-nuclear supernatant, derived from hTERT-RPE1 YFP-Parkin cells treated for 24 h with oligomycin A (1 μM) and antimycin A (1 μM), was incubated with proteinase K in the presence or absence of 1% Triton X-100 for 30 min at 4°C, then analysed by SDS–PAGE and immunoblot. Representative experiment (n = 2).

Source data are available online for this figure.

## USP30 limits basal pexophagy

What then is the role of this distinct peroxisomal pool of USP30? Reversible ubiquitylation has previously been shown to play a key role for the import of peroxisomal matrix proteins via the shuttling receptor PEX5 [33–35]. PEX5 is recycled after import, and this requires transient ubiquitylation by the E3 ligase PEX2. Since the DUB responsible for deubiquitylating PEX5 is currently unknown, in principle this could be a possible role for peroxisomal USP30. Alternatively, and by analogy with its role on mitochondria, USP30 may regulate pexophagy. Peroxisomes have a half life of 1.3–2.2 days, and their turnover can be further regulated in tissues depending on metabolic demand [36,37]. Two recent studies have provided evidence for a possible involvement of PEX2 and ubiquitylation of PEX5, in either ROS- or starvation-induced pexophagy in tissue culture cells [38,39]. The requirement for ubiquitylation in *basal* pexophagy has so far not been directly assessed, but covalent tagging of peroxisomal membrane proteins with ubiquitin has been shown to induce pexophagy [40]. Furthermore, swapping the mitochondrial localisation sequence of USP30 for a peroxisomal membrane targeting sequence was reported to limit basal pexophagy [11].

We generated a pexophagy reporter by tagging the pH-sensitive Keima fluorophore with a peroxisomal matrix targeting sequence (Keima-SKL). Arrival of peroxisomal content in the acidic environment of the lysosome, as confirmed by colocalisation with LAMP1 (Fig EV3A), correlates with an increase in the Em561-signal and a decrease in Em445, which we have false coloured red and green, respectively (Fig 5A, C and D). The Keima-SKL protein is inserted into mature peroxisomes in USP30-depleted and knockout cells, demonstrating that USP30 is not required for peroxisomal import (Fig 5A–D). Strikingly, USP30 depletion with two independent siRNAs resulted in a highly significant increase in the number of red puncta that reflect pexophagy (Fig 5A). This increase in basal pexophagy was contingent on ATG7, indicating the requirement of the canonical autophagy machinery (Figs 5B and EV3B). Reintroduction of wild type but not catalytically inactive, siRNA-resistant USP30 restored pexophagy back to basal levels (Fig 5C). We further corroborated the role for USP30 in basal pexophagy in USP30 CRISPR KO RPE1 cells. These showed an increased level of basal pexophagy that in turn could be suppressed by reintroduction of wild type but not catalytically inactive USP30 (Figs 5D and EV3C). Transfection with mt-Keima, instead of Keima-SKL, showed that these same USP30 KO cells also undergo enhanced basal mitophagy compared to wild-type cells, which likewise could be suppressed by wild type

but not catalytically inactive USP30 (Figs 5E and EV3D). Importantly, analysis of LC3 and p62 in USP30 KO RPE1 cells, treated with an inhibitor of lysosomal acidification (folimycin), showed that our observations do not merely reflect a general increase in autophagic flux (Fig 5F). This conclusion was further corroborated using a well-characterised bulk autophagy reporter (RFP-GFP-rLC3b [41]), monitoring the number of $GFP^+/RFP^+$ autophagosomes and the number of $GFP^-/RFP^+$ puncta as an indicator of autophagic flux (Fig EV3E).

We did not observe any significant change in the number or distribution of catalase-containing mature peroxisomes, nor in the levels of PEX5, PEX19, catalase or PMP70 in USP30-depleted or USP30 KO RPE1 cells (Fig EV3F–I). This suggests that USP30 is not essential for peroxisome maturation and the increased turnover of peroxisomes in these cells may be balanced by an increased rate in biogenesis. Finally, we wondered whether the role of USP30 in basal pexophagy was also linked to PINK1 or Parkin. We noted that expression levels of Parkin are below the detection limit in hTERT-RPE1 cells. Depletion of PINK1 or Parkin had no effect on basal pexophagy, and neither protein was required for the enhancement of pexophagy seen in USP30-depleted cells (Fig 5G).

We have not identified the ubiquitin E3 ligase that USP30 is opposing in this process. Amongst the potential candidates, PEX2 is essential for the biogenesis of mature peroxisomes and thus cannot be easily eliminated. Nevertheless, the fact that only catalytically active USP30 can restore pexophagy to baseline levels, provides the first direct demonstration that basal pexophagy is regulated by ubiquitylation. We also have not pinpointed a specific peroxisomal substrate for USP30, and this may well be challenging without an acute trigger and overexpression of the relevant E3 ligase, both representing artificial conditions we have aimed to avoid in this study. PEX5 has previously been shown to be differentially ubiquity-lated in the context of ROS and starvation-induced pexophagy and peroxisomal matrix protein import [38,39,42,43], yet we have not observed any differentially ubiquitylated species in USP30 knockout cells (Appendix Fig S1).

## A dual role for USP30 in tonically suppressing mitophagy and pexophagy

Taken together, our data suggest a dual role for USP30 in suppressing both basal mitophagy and pexophagy (Fig 5H). Its role in basal mitophagy is linked to PINK1 activity, but is not contingent on amplification by Parkin. Its role in pexophagy is independent of

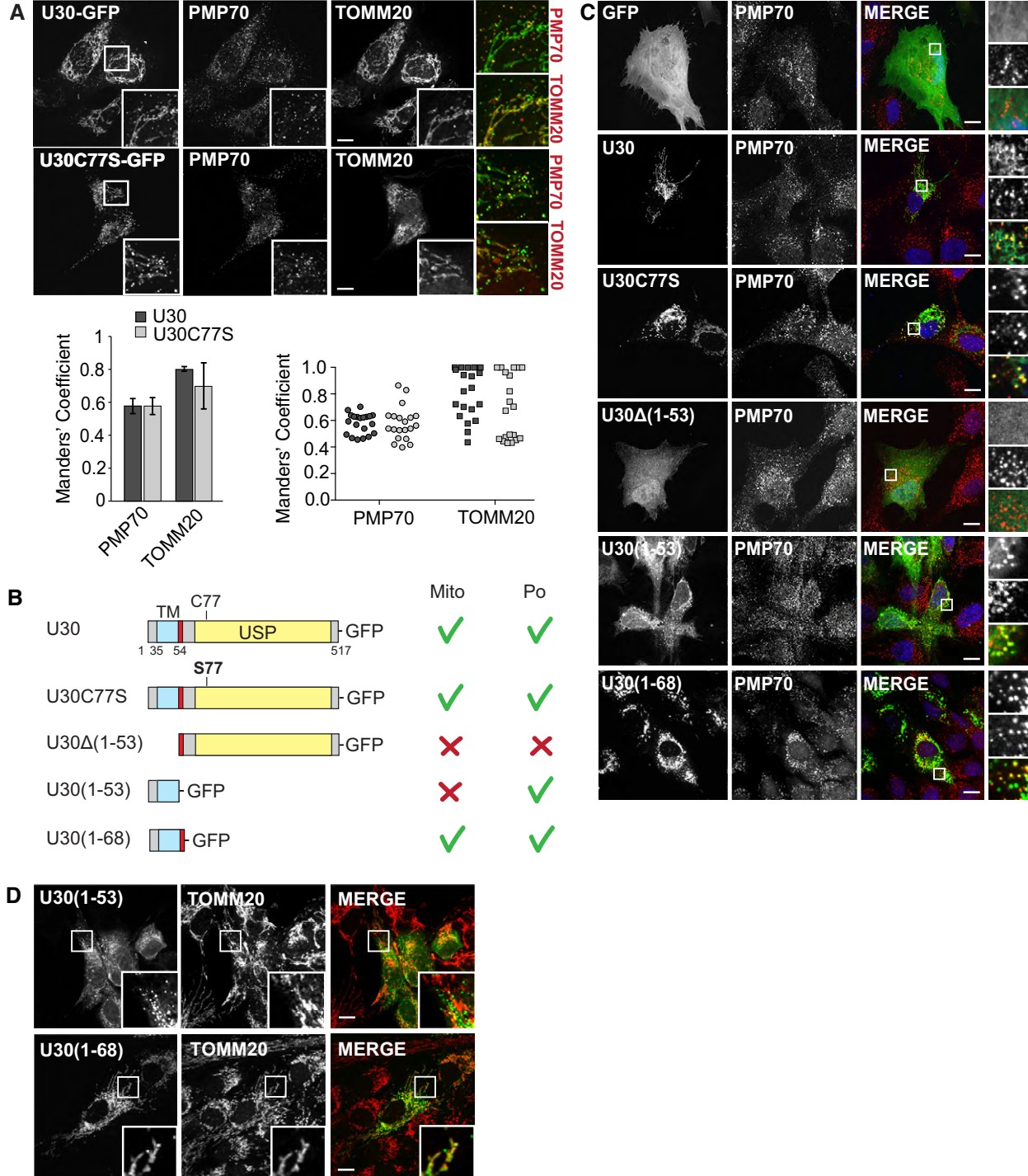

**Figure 4.  Different sequence requirements for mitochondrial and peroxisomal targeting of USP30.**

A  hTERT-RPE1 cells were transfected with USP30-GFP and inactive USP30-C77S-GFP, fixed and co-stained for PMP70 (AlexaFluor555) and TOMM20 (AlexaFluor633). Representative images and quantification. Left graph: Manders' coefficients between USP30 (U30) or USP30-C77S (U30C77S) and PMP70 or TOMM20 derived from z-stacks (average of two independent experiments, ± range). The right hand graph shows all data points.

B  Schematic representation of the USP30 mutants and respective localisation. The transmembrane domain (TM) is indicated in light blue, the catalytic domain (USP) in yellow and the polybasic stretch in red.

C  hTERT-RPE1 cells were transfected with USP30-GFP, USP30-C77S-GFP, USP30Δ(1-53)-GFP, USP30(1-53)-GFP and USP30(1-68)-GFP, fixed and stained for PMP70 (AlexFluor594, red).

D  hTERT-RPE1 cells were transfected with USP30(1-53)-GFP and USP30(1-68)-GFP, fixed and stained for TOMM20 (AlexaFluor594, red). Scale bars (A, C, D) 10 μm.

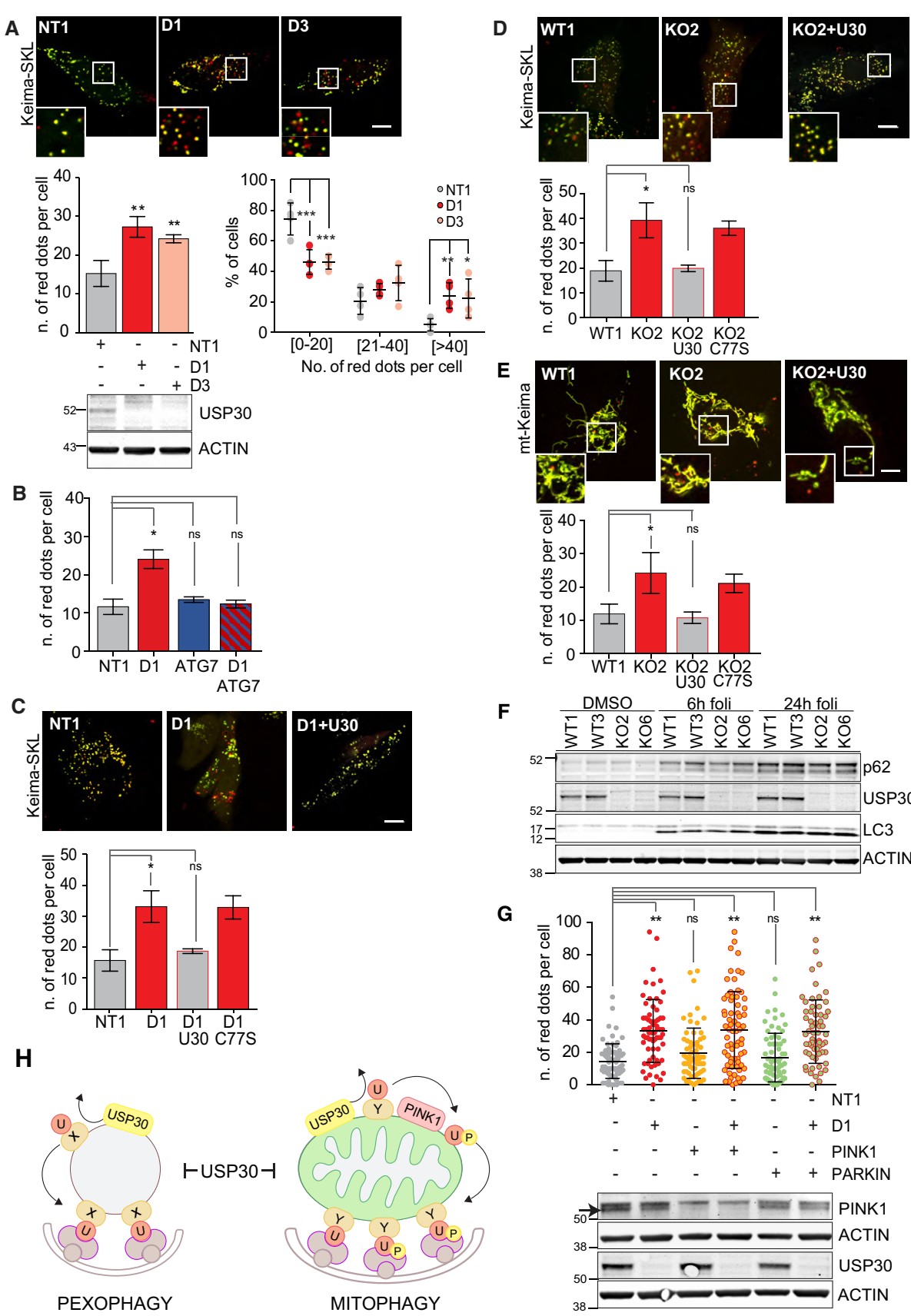

**Figure 5.**

**Figure 5. USP30 restricts basal pexophagy.**

A   Representative images of hTERT-RPE1 cells treated with non-targeting (NT1) or USP30 siRNA oligos (D1, D3). After 24 h, cells were transfected with Keima-SKL and imaged 48 h later. Graphs show the quantification of Keima-SKL "red" puncta (dots) per cell. Average $\pm$ SD, $n$ = 4 independent experiments, 40 cells per experiment. Left: one-way ANOVA and Dunnett's multiple comparison's test, **$P$ < 0.01; right: two-way ANOVA and Bonferroni's multiple comparison test, ***$P$ < 0.001, **$P$ < 0.01, *$P$ < 0.05.

B   Quantification of Keima-SKL as in (A) for hTERT-RPE1 cells treated with siRNA targeting USP30 (D1), ATG7 or both. Average $\pm$ SD, $n$ = 3 independent experiments, 40 cells per experiment; one-way ANOVA and Dunnett's multiple comparison's test, *$P$ < 0.05.

C   Representative images of hTERT-RPE1 cells treated as in (A) except Keima-SKL was cotransfected with siRNA-resistant USP30-GFP (U30) and USP30C77S-GFP (C77S). Three independent experiments were analysed for the first three conditions, two of which also included the C77S rescue. Graph shows the average $\pm$ SD or range; 20 cells per experiment. One-way ANOVA and Dunnett's multiple comparison's test, *$P$ < 0.05.

D   Representative images of Keima-SKL "red" puncta in USP30 KO or WT hTERT-RPE1 cells. Keima-SKL was either transfected on its own or together with USP30-GFP and USP30C77S-GFP for 48 h. Three independent experiments were analysed for the first three conditions, of which two experiments also included the C77S rescue. Graph shows the average $\pm$ SD or range; 20 cells per experiment. One-way ANOVA and Dunnett's multiple comparison's test, **$P$ < 0.01.

E   Representative images of mt-Keima "red"-puncta in USP30 KO and WT hTERT-RPE1 clones, transfected on its own or together with USP30-GFP and USP30C77S-GFP for 48 h with mt-Keima. Graph shows the average from three independent experiments $\pm$ SD; 20 cells per experiment. One-way ANOVA, *$P$ < 0.05.

F   hTERT-RPE1 USP30 WT and KO cells (each two independent clones) were treated for 6 or 24 h with 100 nM folimycin or DMSO as a control. Lysates were separated by SDS–page and analysed by Western blot as indicated.

G   Quantification of Keima-SKL "red" puncta in hTERT-RPE1 cells transfected with siRNA targeting USP30 (D1), PINK1 and PARKIN or non-targeting oligos (NT1). After 24 h, cells were transfected with the Keima-SKL for another 48 h prior to image capture. Average $\pm$ SD, $n$ = 3 independent experiments; 20 cells per experiment. One-way ANOVA and Dunnett's multiple comparison's test, **$P$ < 0.01, overlaid on top of the data points.

H   Summary illustration: USP30 opposes both basal pexophagy- and PINK1-dependent basal mitophagy. Scale bars (A, C, D, E) 10 μm.

Source data are available online for this figure.

PINK1, most likely reflecting the involvement of a distinct set of autophagy receptors. In both pathways, the mechanism of action involves its catalytic activity. It is possible that USP30 activity is directed at a particular set of outer mitochondrial and peroxisomal membrane proteins, which collectively or individually act to nucleate locally restricted, selective autophagy. Previous work records that USP30 shows specificity for Lys6-linked ubiquitin chains and preferentially acts on TOMM20, amongst known Parkin substrates [16,44]. We have not observed significant global changes in the ubiquitylation pattern of USP30 KO lysates or purified mitochondrial fractions (Appendix Fig S2), nor have we seen any changes in TOMM20 ubiquitin status. Given that the fraction of mitochondria undergoing basal mitophagy at any one time is very small, it may be challenging to detect such changes by biochemical analysis of total mitochondrial pools. Importantly, within the context of a slowly progressing pathology like Parkinson's disease, the accumulative effect of a doubling of basal mitophagy could be highly significant.

In summary, we propose a model, which suggests that USP30 maintains the ubiquitylation status of key OMM and peroxisomal membrane proteins at a low level. Thus, USP30 may enable a dynamic ubiquitin economy that is required for multiple core functions at these organelles, whilst preventing inadvertent engagement of the autophagy machinery.

# Materials and Methods

### Cell culture, transfection and RNA interference

Cells were cultured in Dulbecco's modified Eagle's medium DMEM/F12 (hTERT-RPE1, hTERT-RPE1-YFP-PARKIN, hTERT-RPE1-FRT-TREX and SHSY5Y) or DMEM (U2OS, HepG2 and HEK293) supplemented with 10% FBS, 1% non-essential amino acids and 1% penicillin/streptomycin. U2OS cells, stably expressing mCherry-GFP-Fis1(101–152) referred to as U2OS-MGFIS, were a kind gift of Ian Ganley (University of Dundee, [18]), and hTERT-RPE1-FRT-TREX cells were generously donated by Jon Pines (London, [45]). For

siRNA experiments, cells were treated with 40 nM of non-targeting (NT1) or target-specific siRNA oligonucleotides (Dharmacon On-Target Plus or siGenome, ThermoFisher Scientific), using Lipofectamine RNAi-MAX (Invitrogen) according to manufacturer's instructions with a seeding density per well of a six-well plate of $3 \times 10^5$ cells for a 72-h siRNA experiment. Medium was exchanged 6 h after transfection. For depletion of VPS35 in hTERT-RPE1 cells and USP30 in SHSY5Y cells, a "double-hit" protocol was used whereby cells were transfected twice over a 144-h timecourse with a pool of VPS35-targeting siRNAs or an individual USP30 siRNA, respectively. Alternatively, cells were transfected for 24 h with USP30-constructs or 48 h with pcDNA3.1-Keima-SKL or mt-Keima (h)-pIND(SP1) using Genejuice (Novagen). For rescue experiments, pEGFP-N3-USP30-siRes1 or the C77S mutant thereof was transfected together with mt-Keima (h)-pIND(SP1) or pcDNA3.1-Keima-SKL for 48 h.

### siRNA and plasmids

Sequences of siRNAs used in this manuscript were as follows: USP30 siGENOME (D1, 5′-CAAAUUACCTGCCGCACAA-3′; D3, 5′-ACAGGAUGCUCACGAAUUA-3′), ONTARGETplus Non-Targeting siRNA oligo #1 (NT1; 5′-UGGUUUACAUGTCGACUAA-3′), PINK1 (ONTARGETplus pool; 5′-GCAAAUGUGCUUCAUCUAA-3′, 5′-GCUUUCGGCUGGAGGAGUA-3′, 5′-GGACGCUGUUCCUCGUUAU-3′, 5′-GAGACCAUCUGCCCGAGUA-3′), Parkin (ONTARGETplus pool; 5′-GUAAAGAAGCGUACCAUGA-3′, 5′-GAACAUCACUUCAUUACG-3′, 5′-GAUAGUGUUUGUCAGGUUC-3′, 5′-UUAAAGAGCUCCAUCACUU-3′), VPS35 (ONTARGETplus pool; 5′-GAACATATTGCTACCAGTA-3′, 5′-GAAAGAGCATGAGTTGTTA-3′, 5′-GTTGTAAACTGTAGGGATG-3′, 5′-GAACAAATTTGGTGCGCCT-3′), ATG7 (ONTARGETplus pool; 5′-CCAACACACUCGAGUCUU-3′, 5′-GAUCUAAAUCUCAAACUGA-3′, 5′-GCCCACAGAUGGAGUAGCA-3′, 5′-GCCAGAGGAUUCAACAUGA-3′).

The C77S mutation was introduced into USP30 using PCR-based mutagenesis and the following primers: (5′-gttaatttagggaacacAAgcttcatgaactcc-3′, 5′-ggagttcatgaagcTTgtgttccctaaattaac-3′). pEGFP-N3-USP30 and pEGFP-N3-USP30-C77S as well as pEGFP-N3-USP30Δ53,

pEGFP-N3-USP30(1-53), pEGFP-N3-USP30(1-68) and were generated by subcloning the USP30 ORFs from their respective pCR4Topo-entry clones into pEGFP-N3 using restriction enzyme-based ligation. The siRNA-resistant USP30 constructs pEGFP-N3-USP30-siRes1 and pEGFP-N3-USP30C77S-siRes1 were generated by introducing four silent mutations in the region targeted by siRNA D1 (caa atC acA tgT cgG aca aga; mutation in capital) using PCR-based mutagenesis in pCR4Topo-USP30 and C77S, and then, the respective ORFs were subcloned into pEGFP-N3. pRFP-N3-USP30-siRes1, pRFP-N3-USP30-C77S-siRes were generated by replacing EGFP by RFP in the respective pEGFP-N3 clones. All PCR products were sequence-verified and all primer sequences are available on request. mt-Keima (h)-pIND(SP1) was a generous gift from Hiroko Sakurai. To generate pcDNA3.1-Keima-SKL, the Keima sequence from mt-Keima (h)-pIND(SP1) (kind gift from Dr. Atsushi Miyawaki, Riken Brain Science Institute, Japan) was PCR amplified and subcloned into pCDNA3.1 using the following primers: 5′- GCAGGAATTCATGGTGAGCGTGATCGCC-3′ and 5′-GCAGCTC-GAGTTACAGCTTGGTGAACCGCCCAGCAGG-3′. RFP-GFP-rLC3b [41] was kindly provided by Sharon Tooze, London, UK. pCMV6-myc-5′-DDK-mUSP30 was obtained from Origene Technologies Inc, USA.

### Generation of USP30 knockout cells

USP30 knockout cells were generated using CRISPR-Cas9 technology with either one of two USP30 specific sgRNAs targeting exon 3 of isoform 1 (sgRNA1: AGTTCACCTCCCAGTACTCC, sgRNA2: GTCTGCCTGTCCTGCTTTCA). These sgRNAs were cloned into the pSpCas9(BB)-2A-GFP (PX458) vector (a gift from Feng Zhang, Addgene plasmid #48138 [46]) and transfected into hTERT-RPE1-FRT-TREX cells. GFP-positive cells were FACS sorted 24 h after transfection and single cell diluted. Individual clones were amplified and validated by Western blotting and genomic DNA sequencing (sequence available upon request). KO2 used in Fig 5 and Appendix Fig S3 of this manuscript was obtained using sgRNA1, whereas KO6 (Appendix Fig S3) was generated using sgRNA2. WT1 used in Fig 5 and WT3 (see Appendix Fig S3) are clones that underwent the same manipulation with sgRNA1 and sgRNA2, respectively, but retained USP30 expression.

### Antibodies and reagents

Antibodies and other reagents used were as follows: anti-USP30 (Sigma HPA016952, 1:500 for WB), anti-USP30 (gift from Baris Bingol, Genentech [10], 1:100 for IF), anti-PMP70 (Sigma SAB4200181, 1:1,000), anti-PEX19 (Life technologies, PA522129, 1:1,000), anti-catalase (AbCam, ab1877, 1:2,000), anti-PINK1 (Novus Biologicals, BC100-494,) anti-PINK1 (Fig 1E; D8G3, Cell Signalling, 6946S), anti-GFP (gift from Ian Prior, University of Liverpool, Liverpool, UK; 1:5,000), anti-VPS35 (AbCam ab10099, 1:1,000), anti-p62 (BD Transduction, 610833, 1:1,000), anti-LC3 (Nanotools, 5F10, 1:500), anti TOMM20 (Sigma HPA011562, 1:1,000), anti-ACOX1 (gift from T. Hashimoto, Shinshu University, Nagano 390-8621, Japan, 1:10,000), anti-ACOX1 [EPR19038] (Abcam, ab184032, 1:1,000), anti-VDAC1 (AbCam, ab15895, 1:1,000), anti-GSTK1 (SantaCruz, Sc-515580, 1:200), anti ATG7 (Cell Signalling, 12994), mouse anti-actin (AbCam ab6276, 1:10,000), rabbit anti-actin (Sigma A2266, 1:10,000), mouse anti-αtubulin (Sigma T5168, 1:10,000). anti-Myc (Millipore clone 4A6), oligomycin A (SIGMA 75351), antimycin A

(SIGMA A8674), epoxomicin (Millipore 324800) and folimycin (Millipore 344085).

### Preparation cell lysates and Western blot analysis

Cultured cells were lysed with RIPA (10 mM Tris–HCl pH 7.5, 150 mM NaCl, 1% Triton X-100, 0.1% SDS, 1% sodium deoxycholate) or NP-40 (0.5% NP-40, 25 mM Tris–HCl pH 7.5, 100 mM NaCl, 50 mM NaF; only Fig 1E) lysis buffer supplemented with mammalian protease inhibitor cocktail (SIGMA). Proteins were resolved using SDS–PAGE (Invitrogen NuPage gel 4–12%), transferred to nitrocellulose membrane, blocked in 5% milk in PBS or TBS supplemented with Tween-20, and probed with primary antibodies overnight. Visualisation and quantification of Western blots were performed using IRdye 800CW and 680LT coupled secondary antibodies and an Odyssey infrared scanner (LI-COR Biosciences, Lincoln, NE).

### Immunofluorescence and live-cell imaging

Cells were fixed using 4% paraformaldehyde in PBS, permeabilised with 0.2% Triton X-100 in PBS, prior to staining with AlexaFluor-488, 594 or 633 coupled secondary antibodies and imaged using either a Zeiss LSM800 with Airyscan (63× NA 1.4 oil, acquisition software Zen 2.3; Fig 3B and C only), or a 3i Marianas spinning disk confocal microscope (acquisition software Slide Book 3i v3) with a 40× plan-apochromat NA 1.3 oil (Fig 1A) or a 63× plan-apochromat NA 1.4 (all other images) objective and a digital camera (Hamamatsu Flash 4 sCMOS). For live-cell imaging experiments, cells were seeded on a μ-Dish (Ibidi) and images acquired at 37°C in 5% CO$_2$ using a 3i Marianas spinning disk confocal microscope (63× or a 100× oil objective, NA 1.4, Slide Book 3i v3) equipped with a Photometrics Evolve EMCCD camera. The images were further processed using Adobe Photoshop CC2017 and Fiji 1.0 softwares.

All images were acquired sequentially. For colocalisation analysis, z-stacks (10 planes, 3.06 μm range, 0.34 μm step size) of single, double or triple labelled cells were acquired using the 3i Marianas spinning disk confocal microscope (63× oil objective, NA 1.4, Hamamatsu Flash 4 sCMOS camera, Slide Book 3i v3) and analysed with the JaCOP plugin in Fiji using the Costes' automatic threshold to derive Manders' coefficients M1 and M2.

### Analysis of basal mitophagy and pexophagy using fluorescent reporters

For mitophagy quantitation using the U2OS-MGFIS cells, GFP and mCherry live images were acquired sequentially using a 3i Marianas spinning disk confocal microscope (63× oil objective, NA 1.4, Photometrics Evolve EMCCD camera, Slide Book 3i v3). The GFP signal was subtracted from the mCherry signal using ImageJ/Fiji. Resulting images were thresholded, and the number of "red puncta" was determined using the "Analyse particles" function from Fiji.

For pexophagy and mitophagy quantification using Keima-SKL and mt-Keima, respectively, live cells were imaged sequentially (Ex445/Em600, then Ex561/Em600) using 3i Marianas spinning disk confocal microscope (63× oil objective, NA 1.4; Photometrics Evolve EMCCD camera, Slide Book 3i v3). The Ex445/Em600 signal ("green") was subtracted from the Ex561/Em600 signal ("red",

reporting an acidic pH-environment) using ImageJ/Fiji. Resulting images were thresholded, and the number of "red puncta" was determined using the "analyse particles" function in Fiji.

### Subcellular fractionation of HepG2 cells

Subcellular fractionation of HepG2 cells was performed according to a previously established protocol (Manner & Islinger 2017), however using a step gradient instead of a linear gradient for final separation. In brief, HepG2 cells were grown to 80–90% confluency, removed by trypsination, washed once in PBS and resuspended in a homogenisation buffer (HB) consisting of 5 mM MOPS, 250 mM sucrose, 1 mM EDTA, 1 mM DTT, 1 mM ε-aminocaproic acid, 2 mM PMSF, pH 7.4. All further work was carried out on ice or at 4°C. Cells were disrupted by shearing through a syringe with a 27G needle using seven strokes. The resulting homogenate was cleared from debris and undisrupted cells by centrifugation at 600 $g_{av}$ for 10 min. The supernatant was further separated by a differential centrifugation series to produce the heavy mitochondrial pellet (2,000 $g_{av}$, 15 min), the light mitochondrial pellet (20,000 $g_{av}$, 20 min), the microsomal pellet (100,000 $g_{av}$, 30 min) and corresponding cytosolic supernatant. All pellets were resuspended in HB for further Western blot analysis. After removing an aliquot, the light mitochondrial fraction was pelleted (20,000 $g_{av}$, 20 min) and resuspended in 3.0 ml MOPS-buffered (5 mM) Nycodenz solution with a density of 1.15 g/ml. Subsequently, the organelle suspension was layered in between the following MOPS-buffered Nycodenz solutions of a step gradient: 3.0 ml–1.30 g/ml, 3.0 ml–1.17 g/ml, 3.0 ml–1.15 g/ml (organelle suspension), 2.0 ml–1.14 g/ml, 1.5 ml–1.12 g/ml. Centrifugation was performed for 38 min at 100,000 $g_{av}$, 4°C in a VTI50 vertical rotor (Beckman). After the centrifugation, organelles were found to be enriched at the boundaries of the density steps. The gradient was eluted into 6 fractions (L1-L6, Fig 2D and E) corresponding to the visible areas of organelle enrichment. The gradient fractions were diluted at least 3:1 with HB, prior to centrifugation at 100,000 $g$ for 30 min and subsequent resuspension of the pellets in HB for protein quantification. Equal amounts of protein (10 μg/lane) were subjected to SDS–PAGE for immunoblot analysis using organelle-specific antibodies as indicated. After antibody incubation, PVDF membranes were developed with ECL reagent (Thermo Scientific) and chemiluminescence was monitored with a Fusion Solo S instrument (Vilber-Lourmat, Marne-la-Vallée, France).

### Determination of USP30 topology

RPE1-YFP-Parkin cells were homogenised in HIM buffer (200 mM D-mannitol, 70 mM sucrose, 1 mM EGTA, 10 mM HEPES, pH 7.5) by passing four times through a 23 gauge needle. A post-nuclear supernatant (PNS) was obtained by centrifugation at 600 $g$ and subjected to ultracentrifugation at 100,000 $g$ for 40 min at 4°C. Membrane pellets were either resuspended in HIM buffer and analysed directly by SDS–PAGE or extracted in alkaline carbonate buffer (100 mM $Na_2CO_3$, 10 mM Tris pH 11.5 supplemented with mammalian protease inhibitors) or TX100 buffer (2% Triton X-100, 1 M NaCl supplemented with mammalian protease inhibitors) and incubated on ice for 30 min. Samples were then centrifuged again for 40 min at 100,000 $g$ to obtain the pellets and supernatants that were further analysed by SDS–PAGE and immunoblotting.

### Proteinase protection

A PNS was obtained from RPE1-YFP-Parkin cells and either left untreated or incubated with 100 μg/ml proteinase K (Sigma P2308) in the presence or absence of 1% Triton X-100 for 30 min on ice. Samples were then treated with 2 mM PMSF for 5 min on ice prior to analysis by SDS–PAGE and immunoblotting.

### Statistics

*P*-values are indicated as *$P < 0.05$, **$P < 0.01$ and ***$P < 0.001$ and derived either by *t*-test, one or two-way ANOVA, and Dunnett's or Bonferroni's *post hoc* test, respectively, using GraphPad Prism 6.

**Expanded View** for this article is available online.

### Acknowledgements

We thank Jon Lane (University of Bristol, UK), Ian Ganley (University of Dundee, UK) and Jon Pines (ICR, London) for generously providing hTERT-RPE1-YFP-Parkin, MGFIS1-U2OS and RPE1-FlpIN-TREX cells and Atsushi Miyawaki (RIKEN Brain Science Institute, Japan) for the original mt-Keima construct, respectively. We also thank Jacob Corn and Jin-Rui (Amos) Liang for sharing the gRNA sequence used to generate the KO cells, Joe Costello, Marc Fransen and Michael Schrader for sharing reagents and Sharon Tooze and Jon Lane for helpful discussions. We thank David Mason for very helpful suggestions concerning image analysis. This project was funded by a project grant from the Medical Research Council (MR/N00941X/1). JJ is the recipient of a Parkinson's UK studentship (H-1502). EVR-J was funded through a Michael J Fox Foundation Therapeutic Pipeline project grant (13063).

### Author contributions

EM, AM, AK, JJ, EVR-J and SK (Figure 2D–E) were responsible for experimental execution and analysis, with EM taking the lead role for this project. SU, EM, AM, JJ, AK, EVR-J, MI and MJC contributed to the design and analysis of the experiments. SU led, and MJC, MI and EM contributed to, the writing of the manuscript.

### Conflict of interest

The authors declare that they have no conflict of interest.

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
