## [Review Process File · EMBO Reports]

Dual role of USP30 in controlling basal pexophagy and mitophagy

Elena Marcassa, Andreas Kallinos, Jane Jardine, Emma V Rusilowicz-Jones, Aitor Martinez, Sandra Kuehl, Markus Islinger, Michael J Clague, Sylvie Urbé

Review timeline:

Submission date:	5 December 2017
Editorial Decision:	19 January 2018
Revision received:	16 April 2018
Editorial Decision:	14 May 2018
Revision received:	15 May 2018
Accepted:	18 May 2018

Transaction Report:

1st Editorial Decision

19 January 2018

Thank you for the submission of your research manuscript to our journal. We have now received the full set of referee reports that is copied below.

As you will see, while the referees agree that the study is potentially interesting, they also all point out that it requires significant revision before it can be considered for publication here. In particular, the referees point out that a more robust demonstration that USP30 is targeted to peroxisomes and that it functions in pexophagy will be required. Moreover, further data are required to document that USP30 functions in basal mitophagy and that it restricts the basal ubiquitylation of outer membrane proteins.

Upon further discussion with the referees, I would like to add that it is not required to identify substrates of USP30 on peroxisomes. It will however be important to test if PEX5 is ubiquitylated and indeed a USP30 substrate.

Given the overall constructive comments and the potential interest of your findings, we would like to invite you to revise your manuscript with the understanding that the referee concerns must be fully addressed and their suggestions taken on board. Please address all referee concerns in a complete point-by-point response. Acceptance of the manuscript will depend on a positive outcome of a second round of review. It is EMBO reports policy to allow a single round of revision only and acceptance or rejection of the manuscript will therefore depend on the completeness of your responses included in the next, final version of the manuscript.

Revised manuscripts should be submitted within three months of a request for revision; they will otherwise be treated as new submissions. Please contact us if a 3-months time frame is not sufficient for the revisions so that we can discuss the revisions further.

Supplementary/additional data: The Expanded View format, which will be displayed in the main HTML of the paper in a collapsible format, has replaced the Supplementary information. You can submit up to 5 images as Expanded View. Please follow the nomenclature Figure EV1, Figure EV2

etc. The figure legend for these should be included in the main manuscript document file in a section called Expanded View Figure Legends after the main Figure Legends section. Additional Supplementary material should be supplied as a single pdf labeled Appendix. The Appendix includes a table of content on the first page, all figures and their legends. Please follow the nomenclature Appendix Figure Sx throughout the text and also label the figures according to this nomenclature. For more details please refer to our guide to authors.

Regarding data quantification, please ensure to specify the number "n" for how many experiments were performed, the bars and error bars (e.g. SEM, SD) and the test used to calculate p-values in the respective figure legends. Please also include scale bars in all microscopy images.

We now strongly encourage the publication of original source data with the aim of making primary data more accessible and transparent to the reader. The source data will be published in a separate source data file online along with the accepted manuscript and will be linked to the relevant figure. If you would like to use this opportunity, please submit the source data (for example scans of entire gels or blots, data points of graphs in an excel sheet, additional images, etc.) of your key experiments together with the revised manuscript. Please include size markers for scans of entire gels, label the scans with figure and panel number, and send one PDF file per figure or per figure panel.

- a complete author checklist, which you can download from our author guidelines (<http://embor.embopress.org/authorguide#revision>). Please insert page numbers in the checklist to indicate where the requested information can be found.
 - a letter detailing your responses to the referee comments in Word format (.doc)
 - a Microsoft Word file (.doc) of the revised manuscript text
 - editable TIFF or EPS-formatted figure files in high resolution
- (In order to avoid delays later in the publication process please check our figure guidelines before preparing the figures for your manuscript:
http://www.embopress.org/sites/default/files/EMBOPress_Figure_Guidelines_061115.pdf)
- a separate PDF file of any Supplementary information (in its final format)
 - all corresponding authors are required to provide an ORCID ID for their name. Please find instructions on how to link your ORCID ID to your account in our manuscript tracking system in our Author guidelines (<http://embor.embopress.org/authorguide>).

As part of the EMBO publication's Transparent Editorial Process, EMBO reports publishes online a Review Process File to accompany accepted manuscripts. This File will be published in conjunction with your paper and will include the referee reports, your point-by-point response and all pertinent correspondence relating to the manuscript.

I look forward to seeing a revised version of your manuscript when it is ready. Please let me know if you have questions or comments regarding the revision.

REFEREE REPORTS

Referee #1:

Mitochondria and peroxisomes in mammalian cells are selectively turned over by an autophagy pathway, which involves the ubiquitination of their outer membrane proteins. For mitochondria, several deubiquitinating enzymes (DUB) have been reported to negatively regulate mitophagy by

removing the ubiquitin on mitochondria proteins, or on Parkin itself. One such DUB is USP30. However, whether DUBs are involved in pexophagy is not known. Here in this manuscript, Marcassa et al reports that USP30 is not only involved in regulating basal mitophagy but also pexophagy. There are two novel findings in this manuscript. The first regarding mitophagy, they report that USP30 acts upstream of PINK1 by constantly removing ubiquitin on PINK1-substrates. The second is the demonstration that USP30 regulates basal pexophagy in an PINK1/Parkin independent mechanism.

The findings in this manuscript are novel, has broad biological significance and is of importance to the field of selective autophagy. However, there are several major issues with the experimental approach used here that makes it difficult to support the authors' conclusions. One major issue is that the authors use only a single assay to test for either mitophagy or pexophagy. To detect and measure either mitophagy or pexophagy, the authors use a pH-sensitive fluorescent protein to detect the local environment of the protein. The principle of this assay is to detect proteins targeted to lysosomal/autolysosomal compartments by the change in the fluorescence of their FP due to the lower pH of these compartments compared to the cytosol. While this type of assay is widely used in the field, however, other supporting assays must also be present to demonstrate that the localization of their markers to lysosomes are due to mitophagy or pexophagy. This is especially true for pexophagy as this is the first demonstration to link USP30 with pexophagy. A more robust demonstration of USP30 in pexophagy is required. For example, in the USP30 knockdown cells, is the increase in Keima-skl in the lysosomal compartment due to selective autophagy of peroxisomes or increase in overall non-selective autophagy? Does it require ATG5/12/16? Does it also play a role during the activation of pexophagy?

There are several other major concerns that needs to be addressed:

- Another major issue is the use of FIS1 as a mitochondrial targeting protein. It has been well documented that FIS1 targets to both peroxisomes and mitochondria. Therefore, it cannot be used as a marker for mitophagy. For this reason, the calculated 'mitophagy' is likely due to the combination of both peroxisomes and mitochondria in lysosomes. For this reason, the data does not support the idea that USP30 functions upstream of Pink1.
- The principle behind the role of DUBs during the induction of mitophagy is that it counters the activity of a E3 ubiquitin ligase. The authors suggest that USP30 also regulates basal mitophagy and pexophagy. To support this, the authors need to demonstrate that the depletion of USP30 results in an increase in ubiquitin on mitochondria and peroxisomes. This is required to demonstrate that USP30 regulates ubiquitin-mediated mitophagy/pexophagy. This is especially true for pexophagy since less is known about its mechanism.
- The targeting of USP30 to peroxisomes is convincing. However, since most peroxisomal membrane proteins are targeted to peroxisomes via PEX3 and PEX19, the authors need to test whether USP30 targets to peroxisomes through this conical pathway.

Minor issues:

- Figure 2D needs to show MW
- Scale bars are missing in some panels in Fig. 3b

Referee #2:

Summary

In this manuscript, Marcassa and colleagues set out to dissect the role of the deubiquitinase USP30 in the selective autophagy of peroxisomes and mitochondria. Using a pH-sensitive reporter assay, the authors showed that USP30 depletion in unperturbed cells increases mitophagy in a PINK1 dependent manner. As mitochondria and peroxisomes are metabolically linked, the authors examined whether USP30 serve a similar function on peroxisomes. First, the authors found that USP30 indeed partially localizes to peroxisomes under steady state levels and that its peroxisomal targeting is not mediated by mitochondria-derived vesicles. Second, the authors started to map the requirement for peroxisomal and mitochondrial targeting of USP30. Third, the authors established a

pH-sensitive peroxisome reporter and showed that USP30 was required to restrict pexophagy in a PINK1 and PARKIN independent manner. Together, Marcassa et al. provide compelling evidence that USP30 regulates basal autophagic turnover of peroxisomes. This manuscript is elegantly written and all experiments are comprehensibly rationalized and well controlled.

Referee #3:

This manuscript by Marcassa et al. describes the putative role of a deubiquitinating enzyme, USP30, in controlling basal mitophagy and pexophagy in mammalian cells. USP is a mitochondrial outer membrane protein (OMP), that had been shown previously (ref. 10, 12) to counteract PINK1 and PARKIN-mediated mitophagy following acute mitochondrial depolarization by agents like CCCP. Here the authors show the suppression by USP30, of a PINK1-dependent component of basal mitophagy, in cells lacking detectable Parkin. It is proposed that USP30 acts upstream of PINK1 through modulation of ubiquitylated PINK1-substrate availability and thereby affects the initiation of mitophagy. The authors go on to show that a fraction of the endogenous USP30 is independently targeted to peroxisomes where it regulates basal pexophagy in a PINK1- and Parkin-independent manner. The work suggests a role for USP30 in mediating the role of USP30 is removing two major sources of ROS in mammalian cells.

Although a role for USP30 in mitophagy is known, the data come from cells engineered to overexpress large amounts of Parkin and which are subjected to an acute depolarising trigger. Far less is known about the relevance of USP30 function in unperturbed cells expressing limiting amounts of endogenous Parkin. This is the subject of the current study.

The study, while potentially interesting, falls short mechanistically and raises several puzzling questions that limit its impact. In its present state, it is premature for publication.

Depletion of USP30 in U2OS cells reduced basal mitophagy in a PINK1-dependent, but PARKIN-independent manner, whereas PINK1 depletion by itself did not impact basal mitophagy. This phenomenon should be extended to other cell types. It is unclear how the authors conclude that this links USP30 to phospho-ubiquitin.

It is proposed, without much evidence that USP30 restricts the basal ubiquitylation of OMPs, thus limiting PINK1 substrate and in turn recruitment sites for selective autophagy receptors (and Parkin). The target/s of USP30 should be shown experimentally for this model to be credible.

It should be shown that the catalytic activity of USP30 is required for regulation of basal mitophagy.

It should be shown that USP30 is an integral membrane protein of peroxisomes, along with its topology.

The authors suggest that PEX5 is a potential substrate for deubiquitylation by USP30. There is no evidence of this. Additionally, since PEX5 is ubiquitylated at Cys and also at multiple lysines, with different physiological consequences for either biogenesis or pexophagy, it would be important to know whether USP30 acts directly by deubiquitylating K209 in PEX5, which is necessary for (ROS-induced) pexophagy.

The authors claim that this is the first demonstration of ubiquitylation in basal pexophagy, but their effects could be indirect by interfering with ubiquitylation of PEX5 required for biogenesis, and thereby indirectly enhancing basal pexophagy.

Importantly, it is not even clear at this point whether PEX5 or some peroxisomal membrane protein is the target of USP30.

In summary, the most interesting conclusions are very poorly justified making it difficult to recommend publication.

Referee #1:

Mitochondria and peroxisomes in mammalian cells are selectively turned over by an autophagy pathway, which involves the ubiquitination of their outer membrane proteins. For mitochondria, several deubiquitinating enzymes (DUB) have been reported to negatively regulate mitophagy by removing the ubiquitin on mitochondria proteins, or on Parkin itself. One such DUB is USP30. However, whether DUBs are involved in pexophagy is not known. Here in this manuscript, Marcassa et al reports that USP30 is not only involved in regulating basal mitophagy but also pexophagy. There are two novel findings in this manuscript. The first regarding mitophagy, they report that USP30 acts upstream of PINK1 by constantly removing ubiquitin on PINK1-substrates. The second is the demonstration that USP30 regulates basal pexophagy in an PINK1/Parkin independent mechanism.

The findings in this manuscript are novel, has broad biological significance and is of importance to the field of selective autophagy. However, there are several major issues with the experimental approach used here that makes it difficult to support the authors' conclusions. One major issue is that the authors use only a single assay to test for either mitophagy or pexophagy. To detect and measure either mitophagy or pexophagy, the authors use a pH-sensitive fluorescent protein to detect the local environment of the protein. The principle of this assay is to detect proteins targeted to lysosomal/autolysosomal compartments by the change in the fluorescence of their FP due to the lower pH of these compartments compared to the cytosol. While this type of assay is widely used in the field, however, other supporting assays must also be present to demonstrate that the localization of their markers to lysosomes are due to mitophagy or pexophagy. This is especially true for pexophagy as this is the first demonstration to link USP30 with pexophagy. A more robust demonstration of USP30 in pexophagy is required. For example, in the USP30 knockdown cells, is the increase in Keima-skl in the lysosomal compartment due to selective autophagy of peroxisomes or increase in overall non-selective autophagy? Does it require ATG5/12/16? Does it also play a role during the activation of pexophagy?

We thank the reviewer both for the recognition that our findings are novel and biologically significant as well as for their insightful questions and suggestions.

We recognise that our assessment of basal mitophagy and pexophagy is based on a single type of assay and that it would be preferable to have orthogonal approaches on hand to back this up. However, we are not aware of another approach that would be of equal sensitivity – ie allow us to pick out a two fold increase of the basal turnover of mitochondria and peroxisomes that does not affect the overall number of peroxisomes. It is inherently difficult to monitor a low level stochastic event using a biochemical approach. These highly sensitive imaging read-outs allow us to single out and isolate these events on a single cell basis. We have mitigated this fact by using two different mitophagy reporters and in response to the valid comments with regards to Fis1 (discussed below), have added additional experiments based on the mt-keima probe (**New Fig. 1E**, see below, response to reviewer 3).

• ...is the increase in Keima-skl in the lysosomal compartment due to selective autophagy of peroxisomes or increase in overall non-selective autophagy?

This also had occurred to us and we had already included a western blot showing that USP30 deletion affects neither steady state LC3I/II nor p62 levels nor flux as measured \pm folimycin, an inhibitor of the vacuolar ATPase, suggesting that overall bulk (non-selective) autophagy and autophagic flux are not affected (Old Fig. EV3C). We have now moved this Figure into the Main Figure section (**New Fig. 5F**). We have also conducted corresponding experiments in USP30 siRNA treated cells further supporting this point (see **Rebuttal Figure 1**). Finally, we have added an additional

analysis of a commonly used tandem tag autophagy reporter, RFP-GFP-LC3 (**New Fig. EV3F**, [1]). Whilst we have shown that both mt-keima and keima-SKL report increased basal mitophagy and pexophagy in our USP30 KO cells, we do not find any concomitant increase in the number of RFP+/GFP+ LC3 punctae (ie number of autophagosomes) nor in the number of RFP+/GFP- LC3 punctae, which report on autophagic flux.

• Does it require ATG5/12/16? Does it also play a role during the activation of pexophagy?

Very little is known regarding the mechanism of basal peroxisome turnover in human cells. The two most commonly used acute triggers of pexophagy in human cells, for which some of the molecular mechanisms have been proposed, namely hydrogen peroxide treatment and amino acid starvation, are both triggers of global autophagy. It is unclear whether the molecular mechanisms underlying basal pexophagy are the same or differ from those at play in H₂O₂ treated or starved cells. In addition, H₂O₂ treatment would be expected to inactivate USP30 through oxidation of the catalytic cysteine, thus under these conditions the presence or absence of USP30 may be irrelevant. For these reasons, we have currently limited our analysis to basal levels of pexophagy – ie the basal turnover of peroxisomes.

In order to further characterise the basal pexophagy events we capture with our approach, we now also show that our “red (Em561)” Keima-SKL puncta colocalise with the lysosomal marker protein LAMP1 (**New Fig. EV3A**). Importantly, as suggested by this reviewer, we have also asked the question whether the canonical autophagy machinery is required for the basal pexophagy events we monitor. To this end, and upon consultation with several colleagues who are autophagy experts (Sharon Tooze, Jon Lane), we have used siRNA to deplete ATG7, rather than ATG5, either on its own or together with USP30. Our results (**New Fig. 1C and 5B**) show that the enhancement of both basal mitophagy and pexophagy we see upon USP30 depletion is dependent on ATG7. Thus we conclude that a component of the core autophagy machinery is required for both processes.

There are several other major concerns that needs to be addressed:

• Another major issue is the use of FIS1 as a mitochondrial targeting protein. It has been well documented that FIS1 targets to both peroxisomes and mitochondria. Therefore, it cannot be used as a marker for mitophagy. For this reason, the calculated 'mitophagy' is likely due to the combination of both peroxisomes and mitochondria in lysosomes. For this reason, the data does not support the idea that USP30 functions upstream of Pink1.

We agree with the reviewer that this is a potential caveat. FIS1 (like USP30) is indeed a protein that has been reported to target to both mitochondria and peroxisomes. However, most mammalian cell lines including U2OS cells have a low amount of peroxisomes such that we are confident that the bulk of FIS1 is localised to mitochondria. Furthermore the fact that we see a clear dependence on PINK1 for USP30 induced mitophagy in the MGFIS-U2OS cells, whilst we do not find a role for PINK1 in pexophagy, fits with the idea that we are primarily monitoring mitophagy in these cells.

In our first submission, we also reported increased mitophagy in USP30 KO versus wild-type RPE1 cells using an alternative, widely-used, matrix targeted selective mitophagy reporter mt-keima (**Fig. 5E**). We now also show that this result is recapitulated in USP30 siRNA depleted cells and importantly remains contingent on the presence of PINK1 (**New Fig. 1E**).

• The principle behind the role of DUBs during the induction of mitophagy is that it counters the activity of a E3 ubiquitin ligase. The authors suggest that USP30 also regulates basal mitophagy and pexophagy. To support this, the authors

need to demonstrate that the depletion of USP30 results in an increase in

ubiquitin on mitochondria and peroxisomes. This is required to demonstrate that USP30 regulates ubiquitin-mediated mitophagy/pexophagy. This is especially true for pexophagy since less is known about its mechanism.

We maintain that the fact that wild-type but not catalytically inactive USP30 can revert the enhanced basal mitophagy and pexophagy we observe in USP30 knockdown/knockout cells, is in itself firm evidence that (de-)ubiquitylation is involved in the process.

By focusing on basal levels of mitophagy and pexophagy, we are looking here at rare stochastic events. In contrast to many other studies, we are not over-expressing an Ubiquitin E3 ligase. The assessment of the keima fluorophore in acidic lysosomes is an accumulative end-point measurement, that reflects the turnover of a minor fraction of bulk mitochondria or peroxisomes. Thus the global basal ubiquitylation levels which are detected biochemically may not inform on the relevant organellar sub-population.

Two other points are relevant here which we now also include in the revised text. USP30 shows specificity for relatively rare K6 Ubiquitin linkages and even then does not act universally on all such mitochondrial substrates modified by activated Parkin [2, 3].

We have modified our description (text and legend) of the model (Fig. 1F) proposed in our original submission to reflect these nuances. Fig. 5H merely serves to illustrate the dual role of USP30 on the two organelles.

We have carried out a variety of experiments to compare ubiquitylation in RPE1 wild-type versus KO USP30 cells without detecting any significant differences: We have probed cell lysates for total ubiquitin as well as pulled down poly-ubiquitin chains using TUBES and analysed the samples using both total Ubiquitin antibody as well as K6-Ub-specific affimers, without detecting a significant difference in global ubiquitylation levels (**Appendix Fig.2A**). We have also probed mitochondria enriched fractions and purified mitochondria from wild-type and KO RPE1 cells using a mitochondrial purification kit (MACS Miltenyi Biotech) for total ubiquitin and again did not observe significant differences (**Appendix Fig.2D**). Likewise, a 20000 g membrane fraction containing the bulk of peroxisomes did not reveal any distinctive changes in ubiquitylation in USP30 KO cells (**Appendix Fig.1E**).

We conclude that it is most likely not a global change in ubiquitylation status but rather a particular set of “trigger” proteins that accumulate in a ubiquitylated form upon USP30 depletion, and which would be most relevant at sites of PINK1 accumulation, ie in proximity of the TOMM-complex. We have not observed any changes in the ubiquitylation status of TOMM20 in our USP30 KO RPE1 cells (**Appendix Fig. S2B and C, Rebuttal Fig. 2**), thus it is possible that TOMM20 is a primary substrate of USP30 only in the context of Parkin expressing cells.

With regards to pexophagy, most cell lines, including our RPE1 WT and USP30 KO cells, do not contain a large amount of peroxisomes, which are thus difficult to purify away from mitochondria. This, together with the fact that the bulk of USP30 is associated with mitochondria, makes it impossible to attribute alterations in global ubiquitylation patterns seen in peroxisomal fractions unequivocally to peroxisomal proteins.

As requested by the reviewers and editor, we have probed extensively for PEX5, as a peroxisomal protein that has previously been shown to be ubiquitylated. We have not detected any differential, distinctive ubiquitylation pattern of PEX5 in our USP30 knockout cells 1) comparing PEX5 by western blotting under reducing and nonreducing conditions, previously shown to reveal the Cys-conjugated monoubiquitylated PEX5 that is implicated in peroxisomal import, 2) analysing total lysates from untreated, H₂O₂ and epoxomycin treated cells, 3) immunoprecipitating endogenous PEX5 and probing for ubiquitin in untreated cells or in cells treated with

H2O2 or epoxomycin (see **Appendix Fig. 1**).

We are planning to analyse global ubiquitin proteomics in different USP30 KO cell lines, in order to comprehensively identify direct substrates of USP30 – however we believe that this analysis goes beyond the remit of this study.

• The targeting of USP30 to peroxisomes is convincing. However, since most peroxisomal membrane proteins are targeted to peroxisomes via PEX3 and PEX19, the authors need to test whether USP30 targets to peroxisomes through this canonical pathway.

We are pleased that this reviewer is satisfied with our data showing that USP30 is targeted to peroxisomes. We notice that the minimal peroxisomal targeting sequence of USP30, which includes a short transmembrane domain, is very well compatible with a PEX19 mediated targeting mechanism. In addition, preliminary experiments indicate that USP30-GFP can interact with endogenous PEX19 and requires its transmembrane domain to do so (**Rebuttal Figure 3**).

Unfortunately, efficient depletion of PEX19 results in a defect in peroxisome biogenesis, making this a difficult point to answer in a direct fashion. This is by far the most likely pathway for USP30 insertion, but to prove this would require a series of non-trivial *in vitro* experiments to reconstitute insertion into the peroxisomal membrane with *in vitro* translated protein. In response to a query from reviewer 3, we now show that peroxisomal USP30 is an integral membrane protein with its catalytic domain exposed to the cytosol. Interestingly, USP30 is not a tail-anchored (TA) protein, in contrast to other well characterised dually targeted mitochondrial and peroxisomal proteins. We have added a statement to this effect, but would rather not speculate on the association of USP30 with PEX19 at this stage.

Minor issues:

- **Figure 2D needs to show MW**
- **Scale bars are missing in some panels in Fig. 3b**

Thank you for pointing this out. We have amended these issues.

Referee #2:

Summary

In this manuscript, Marcassa and colleagues set out to dissect the role of the deubiquitinase USP30 in the selective autophagy of peroxisomes and mitochondria. Using a pH-sensitive reporter assay, the authors showed that USP30 depletion in unperturbed cells increases mitophagy in a PINK1 dependent manner. As mitochondria and peroxisomes are metabolically linked, the authors examined whether USP30 serve a similar function on peroxisomes. First, the authors found that USP30 indeed partially localizes to peroxisomes under steady state levels and that its peroxisomal targeting is not mediated by mitochondria-derived vesicles. Second, the authors started to map the requirement for peroxisomal and mitochondrial targeting of USP30. Third, the authors established a pH-sensitive peroxisome reporter and showed that USP30 was required to restrict pexophagy in a PINK1 and PARKIN independent manner. Together, Marcassa et al. provide compelling evidence that USP30 regulates basal autophagic turnover of peroxisomes. This manuscript is elegantly written and all experiments are comprehensibly rationalized and well controlled.

We are very pleased that this reviewer likes and approves of our manuscript.

Referee #3:

This manuscript by Marcassa et al. describes the putative role of a deubiquitinating enzyme, USP30, in controlling basal mitophagy and pexophagy in mammalian cells. USP is a mitochondrial outer membrane protein (OMP), that had been shown

previously (ref. 10, 12) to counteract PINK1 and PARKIN-mediated mitophagy following acute mitochondrial depolarization by agents like CCCP. Here the authors show the suppression by USP30, of a PINK1-dependent component of basal mitophagy, in cells lacking detectable Parkin. It is proposed that USP30 acts upstream of PINK1 through modulation of ubiquitylated PINK1-substrate availability and thereby affects the initiation of mitophagy. The authors go on to show that a fraction of the endogenous USP30 is independently targeted to peroxisomes where it regulates basal pexophagy in a PINK1- and Parkin-independent manner. The work suggests a role for USP30 in mediating the role of USP30 is removing two major sources of ROS in mammalian cells.

Although a role for USP30 in mitophagy is known, the data come from cells engineered to overexpress large amounts of Parkin and which are subjected to an acute depolarising trigger. Far less is known about the relevance of USP30 function in unperturbed cells expressing limiting amounts of endogenous Parkin. This is the subject of the current study.

The study, while potentially interesting, falls short mechanistically and raises several puzzling questions that limit its impact. In its present state, it is premature for publication.

We thank this reviewer for a concise summary of our study and address the specific concerns below.

• **Depletion of USP30 in U2OS cells reduced basal mitophagy in a PINK1-dependent, but PARKIN-independent manner, whereas PINK1 depletion by itself did not impact basal mitophagy. This phenomenon should be extended to other cell types.**

USP30 depletion *increases* basal mitophagy.

The fact that PINK1 depletion does not affect basal levels of mitophagy may at first be surprising. However, whilst our manuscript has been under review, in agreement with our assessment, Ian Ganley's laboratory has reported that basal mitophagy observed in the Mito-QC mouse (expressing the MGFIS-reporter) is not affected by PINK1 KO (McWilliams et al., Cell Metabolism 2018). In addition, in collaboration with Alex Whitworth and colleagues, we have recently reported a lack of impact of PINK1 knockout on basal mitophagy in flies (Lee et al., J. Cell Biol. 2018 in press). Neither of these studies have assessed the role of USP30 in either of these settings.

In addition to the results we present in U2OS-MGFIS cells in Figure 1, we have already shown the effect of USP30 KO on hTERT-RPE1 cells expressing mt-keima in Figure 5E. We have now extended this observation in a USP30 siRNA mediated depletion setting and assessed sensitivity to concomitant PINK1 depletion (**New Fig. 1E**). Also in RPE1 cells, which likewise do not express detectable levels of Parkin, basal mitophagy is independent of PINK1 whilst the enhanced mitophagy seen upon USP30 depletion depends on PINK1.

• **It is unclear how the authors conclude that this links USP30 to phosphoubiquitin.**

The link to phospho-Ubiquitin comes from the fact that Ubiquitin is now recognised as the key substrate for PINK1 in mitophagy. Our model is a working hypothesis compatible with our epistatic findings. We have further modified the text to clarify this point.

• **It is proposed, without much evidence that USP30 restricts the basal ubiquitylation of OMPs, thus limiting PINK1 substrate and in turn recruitment sites for selective autophagy receptors (and Parkin). The target/s of USP30 should be shown experimentally for this model to be credible.**

We thank the reviewer for this important point. We have revised our model to reflect nuances we had previously omitted but which are discussed in the revised text (see our response to reviewer 1). Our figure is highlighted as a model rather than summary of findings and as such seeks to provide the simplest interpretation of our results, whilst stimulating further work. We have detailed proteomics

experiments planned with our panel of knockout cells, which we will report on at a later point. Note that the question of the most relevant or primary substrate is an issue that is still hotly debated for Parkin-dependent mitophagy. We reiterate that the requirement for catalytic activity implicates deubiquitylation in the process.

• It should be shown that the catalytic activity of USP30 is required for regulation of basal mitophagy.

We now show that catalytic activity of USP30 is indeed required for the regulation of basal mitophagy (New Figure 5E).

• It should be shown that USP30 is an integral membrane protein of peroxisomes, along with its topology.

We thank the reviewer for this suggestion although this is not a straight forward experiment: Firstly, pure peroxisomal fractions are not easy to produce to enable biochemical *in vitro* experiments that rely on retaining organelle membrane integrity. Secondly, in mammalian cells peroxisomes make up a low percentage of all total membranes and thus in such an experiment it will be near impossible to be sure that we are analyzing peroxisomal and not mitochondrial USP30. To tackle this issue we have made use of our YFP-Parkin overexpressing cells in which we can deplete mitochondria upon acute depolarization. Under these conditions we have examined the distribution of endogenous USP30. USP30 is clearly resistant to Na₂CO₃ extraction and the C-terminal portion, which we can detect with a specific antibody, is accessible to Proteinase K treatment. These data which have now been included as New Figures 3C and D in the revised manuscript, support the notion that the peroxisomal pool of USP30 is an integral transmembrane protein with its C-terminus exposed to the cytosol.

Note that we could not use differentially tagged proteins for these experiments as Nterminal tagging of USP30 renders the protein cytosolic. We have also shown that the transmembrane domain of USP30 is required for its targeting to peroxisomes.

• The authors suggest that PEX5 is a potential substrate for deubiquitylation by USP30. There is no evidence of this. Additionally, since PEX5 is ubiquitylated at Cys and also at multiple lysines, with different physiological consequences for either biogenesis or pexophagy, it would be important to know whether USP30 acts directly by deubiquitylating K209 in PEX5, which is necessary for (ROS-induced) pexophagy.

We believe the reviewer may have misinterpreted our statements – in the discussion we clearly state: “We also have not pinpointed a specific peroxisomal substrate for USP30 and this may well be challenging without an acute trigger and overexpression of the relevant E3 ligase, both representing artificial conditions we have aimed to avoid in this study.” The only statement we made with respect to PEX5 as a substrate for USP30 was in the introduction to the peroxisome section where we proposed PEX5 as a possible substrate in the context of import: “PEX5 is recycled after import and this requires transient ubiquitylation by the E3-ligase PEX2. Since the DUB responsible for deubiquitylating PEX5 is currently unknown, this could be a possible role for peroxisomal USP30.” We have now changed this statement to: “...this could, in principle, be a possible role for peroxisomal USP30.” Importantly, we demonstrate that USP30 depletion does not affect import or maturation of peroxisomes since we can incorporate Keima-SKL and we do not see any significant changes in catalase/PMP70 colocalisation. We have clarified our statements regarding this point.

As this reviewer rightly points out, PEX5 ubiquitylation has been linked not just to matrix protein import but also to pexophagy. Ubiquitylation of PEX5 on K209 has been implicated in pexophagy, although this has only been demonstrated in cells overexpressing the ubiquitin E3 ligase PEX2 and in response to large doses of hydrogen peroxide. USP30 like other members of the family depends on highly reactive cysteine residues and is thus expected to be inactivated under these

conditions. Moreover in the cells used in this study, we have struggled to detect any distinctive K-linked mono-ubiquitylation of PEX5 except upon inhibition of the proteasome.

As requested by the reviewers and editor, we have probed extensively for PEX5, as a peroxisomal protein that has previously been shown to be ubiquitylated. We have not detected any differential, distinctive ubiquitylation pattern of PEX5 in our USP30 knockout cells 1) comparing PEX5 by western blotting under reducing and nonreducing conditions, previously shown to reveal the Cys-conjugated monoubiquitylated PEX5 that is implicated in peroxisomal import, 2) analysing total lysates from untreated, H₂O₂ and epoxomycin treated cells, 3) immunoprecipitating endogenous PEX5 and probing for ubiquitin in untreated cells or in cells treated with H₂O₂ or epoxomycin (see **Appendix Fig. 1**).

In summary, we have found no evidence to suggest that PEX5 is a substrate for USP30. We have not seen any changes in total levels, nor any appearance of higher molecular weight species of PEX5 in USP30 depleted or USP30 KO cells, in any of the cell types we have studied.

• The authors claim that this is the first demonstration of ubiquitylation in basal pexophagy, but their effects could be indirect by interfering with ubiquitylation of PEX5 required for biogenesis, and thereby indirectly enhancing basal pexophagy.

We also wondered about this, however as we have indicated in the original manuscript, and state above, if we interfered with biogenesis we would not be able to import Keima-SKL in the knockdown or KO cells (Old Figure 5A-C). We also would predict that an effect on biogenesis would have resulted in a marked decrease in peroxisomes staining positive for catalase – this is not the case (Figure E3H-K). Finally, since we monitor Keima-SKL positive peroxisomes, we do monitor the turnover of mature peroxisomes. We have further clarified these issues in the revised manuscript.

• Importantly, it is not even clear at this point whether PEX5 or some peroxisomal membrane protein is the target of USP30.

We agree with this assessment and have stated this clearly in the discussion of the manuscript.

• In summary, the most interesting conclusions are very poorly justified making it difficult to recommend publication.

We thank the reviewer for recognising the interest of our conclusions to the field.

These are now more firmly supported by the new data we have added.

Not every avenue has reached a neatly boxed off conclusion, but we believe we discuss any limitations in a balanced manner. We have more clearly highlighted the speculative nature of our model (Figure 1F), but would suggest that such models can help stimulate thinking in the field.

1. Kimura S, Noda T, Yoshimori T (2007) Dissection of the autophagosome maturation process by a novel reporter protein, tandem fluorescent-tagged LC3. *Autophagy* **3**: 452-460.
2. Gersch M, Gladkova C, Schubert AF, Michel MA, Maslen S, Komander D (2017) Mechanism and regulation of the Lys6-selecting deubiquitinase USP30. *Nat Struct Mol Biol* **24**: 920-930.
3. Sato Y et al (2017) Structural basis for specific cleavage of Lys6-linked polyubiquitin chains by USP30. *Nat Struct Mol Biol* **24**: 911-919

2nd Editorial Decision

14 May 2018

Thank you for the submission of your revised manuscript to EMBO reports. I apologize again for the delay in handling your manuscript. I have sent the revised version back to referees #1 and #3. Unfortunately, referee 1 was not available anymore to assess the revised version. Since I wanted to

avoid the inclusion of a fourth, new referee, I have therefore asked an editorial advisor with expertise in mitophagy and ubiquitin to step in as advisor and to assess how well you have responded to the concerns of this referee and to advise if the revised version is suitable for publication. Please find the comments from referee #3 and the advisor (referee #4) below.

As you will see, the opinions remain divided. While the advisor (and former referee 2) support publication, referee 3 remains concerned that the effects of USP30 on mitophagy and pexophagy are small. I have discussed this further with the advisor and you have also provided feedback on these concerns. Based on this feedback, we have decided to allow you to submit a final revision and to address the remaining points in the text and discussion.

Please make sure to clearly state in the paper that the difference in mitophagy is only 2-fold and avoid any overstatements.

Regarding the second point, i.e., the effect of USP30 acting upstream of PINK1 our advisor noted that "... the authors have a reasonable point by saying that USP30 can modulate PINK1 substrate availability" and suggests to explain this clearly in the text and discussion, which you anyway might want to update to include recent work on PINK1-independent mitophagy.

Also the third point, i.e., the effect on peroxisomes and peroxisomal proteins, should be addressed and commented upon in the discussion, as appropriate.

Please also provide a point-by-point response to the referee comments.

Browsing through the manuscript myself, I noticed a few editorial things that we need before we can proceed with the official acceptance of your manuscript.

REFEREE REPORTS

Referee #3:

I reiterate my previous opinion that this manuscript falls significantly short of the journal standards for publication. The effects of USP30 on mitophagy and basal pexophagy are small (generally less than 2 fold as the authors admit, but do not state explicitly and consistently throughout the manuscript), the requirement for DUB activity is very weak at best (Fig. 5E) and no effects are seen on peroxisomal proteins necessary for pexophagy.

I still do not understand how the USP30 effect on basal mitophagy could be PINK1 dependent, but depletion of PINK1 does not affect basal mitophagy. The epistasis evidence that USP30 acts upstream of PINK1 is weak and the statement that "USP30 acts upstream of PINK1 through modulation of PINK1 substrate availability" is speculative.

The statement that "We did not observe any significant change in the number or distribution of catalase-containing mature peroxisomes, nor in the levels of PEX5, PEX19, Catalase or PMP70 in USP30 depleted or USP30 KO RPE1 cells (Fig. EV3F-I), suggests that the effect on pexophagy is really weak and not reflected by changes in these peroxisomal matrix and membrane proteins.

Among other related points, the evidence for the integral membrane property is not convincing because both PMPs and catalase (a peroxisomal matrix protein) are largely carbonate insoluble (Fig. 3C).

This leaves us with a weak phenomenological observation of small effects of mitophagy and pexophagy that comes with no mechanistic insight. No target of USP30 is shown for either mitophagy or pexophagy. The model presented may be an explanation of the data, but that does not mean it is correct.

Referee #4:

Ubiquitination was previously established as an important signal that regulates selective autophagy. Marcassa and colleagues report now that USP30, a deubiquitinating enzyme, is involved in regulation of basal mitophagy and pexophagy pathways. USP30 is localized in both organelles and requires its DUB activity for the effect on autophagy pathways. In the case of mitophagy, the novel finding is that USP30 acts upstream of PINK1 and trims Parkin-ubiquitinated substrates during basal mitophagy responses. On the other side, USP30 regulates basal pexophagy independent of PINK1 and Parkin pathways. They also show that the enhancement of both basal mitophagy and pexophagy upon USP30 depletion is dependent on ATG7 indicating that the core autophagy machinery is required for both processes. Importantly they have shown that wild type USP30 but not DUB inactive USP30 can revert the enhanced basal mitophagy and pexophagy upon USP30 knockdown. These are all novel and very important findings in the field of selective autophagy. On a technical side the revised version has offered several new approaches to address most of previously raised valid issues. In cases when there was no better way or approach the authors have provided additional supporting evidence. As such I find this work suitable for publication in EMBO based on both novelty/advance in the field as well as adequate technical quality of assays.

2nd Revision - authors' response

15 May 2018

Referee #3

I reiterate my previous opinion that this manuscript falls significantly short of the journal standards for publication. The effects of USP30 on mitophagy and basal pexophagy are small (generally less than 2 fold as the authors admit, but do not state explicitly and consistently throughout the manuscript), the requirement for DUB activity is very weak at best (Fig. 5E) and no effects are seen on peroxisomal proteins necessary for pexophagy.

We respectfully, but strongly, disagree with this assessment.

Firstly, regarding the magnitude of the effects we see, USP30 depletion *and* knockout consistently result in an approximately two-fold increase in mitophagy and pexophagy respectively. This is clearly stated for mitophagy in the manuscript on page 5, 1st paragraph, and otherwise very easily deducible from the figures. This difference is highly reproducible and significant. Our paper deals with basal mitophagy and pexophagy in distinction to most studies of these pathways, which use over-expression of ubiquitin E3 ligases (eg Parkin or PEX2 respectively) and an acute perturbation (eg mitochondrial depolarisation, ROS or starvation). The absolute levels of mitophagy and pexophagy we are dealing with are thus lower and this offers technical challenges. However, we fail to see how a two-fold change can be considered small. Consider for example a two fold change in cell volume or, to choose a different context, a two fold change in the world population, neither of which could reasonably be considered a small change. Within the context of a slowly progressing pathology like Parkinson's disease that results from the buildup of damage over decades, the accumulative effect of a doubling of basal mitophagy could be highly significant. We have added this latter contextual argument to the discussion on page 10.

Secondly, the requirement for DUB activity is very solid and we are puzzled by the stance of this referee vis-a-vis our data (Figure 5C, D and E).

Thirdly, we are not surprised that we did not see any global changes in the amount of peroxisomal proteins as we are not triggering global pexophagy, but merely monitoring the probability of a rare and stochastic event. The assessment of the keima fluorophore in acidic lysosomes is an accumulative end-point measurement, that reflects the turnover of a minor fraction of bulk peroxisomes. In other words, only a small percentage of the peroxisomes undergo pexophagy at any one time, thus even a two fold increase in the rate does not have a major impact on steady state levels of peroxisomal proteins. Furthermore, as we have already stated in the text (page 18), it is possible that the increased turnover of peroxisomes in USP30 knockout cells is balanced by a corresponding increased rate in biogenesis.

I still do not understand how the USP30 effect on basal mitophagy could be PINK1 dependent, but depletion of PINK1 does not affect basal mitophagy. The epistasis evidence that USP30 acts upstream of PINK1 is weak and the statement that "USP30 acts upstream of PINK1 through modulation of PINK1 substrate availability" is speculative.

Our observations, showing that basal levels of mitophagy are not affected by PINK1 depletion, are corroborated, as laid out in our first rebuttal letter and also discussed in our manuscript (p5, third line), by two recent studies using fluorescent mitophagy reporters in mice (Mito-QC mouse, McWilliams et al., *Cell Metabolism* 2018) and in flies (Lee et al., *J. Cell Biol.* 2018). This suggests that the major component of mitophagy is independent on PINK1 in multiple systems and may also be independent on ubiquitin.

Our results show that USP30 depletion can only enhance mitophagy in cells that do express PINK1. Our interpretation of these epistatic observations in two cell lines, with two distinct reporter systems is clearly signposted as a working model that seeks to provide the simplest interpretation of our results, whilst stimulating further work.

We suggest that depletion or knockout of USP30 reveals a distinct component of PINK1-dependent basal mitophagy that is tonically suppressed by USP30. As we have stated in the discussion, this tonic suppression may prevent inadvertent engagement of the autophagy machinery whilst allowing the system to be held at a hair trigger. The sensor of this system, PINK1, has a high turnover rate, which is contingent on its engagement with the mitochondrial import system. Thus PINK1 continuously surveys the mitochondrial environment. There are many instances of this level of control in biology; consider for example the role of the tumour suppressor PTEN in limiting basal PtdInsP₃ levels and thereby restricting signaling.

We propose that USP30, as an interacting partner of TOMM20 is positioned in close proximity to the outer mitochondrial import channel. The key signal that initiates the mitophagy cascade is the phosphorylation of ubiquitin by PINK1 at the outer mitochondrial membrane (OMM), which can then activate and stabilise Parkin at the OMM. It is currently unclear, which protein or proteins the ubiquitin that is targeted by PINK1 is attached to, and which ubiquitin E3 ligase is acting as a primer in this cascade. Candidates that are either associated with mitochondria and/or have already been shown to ubiquitylate OMM proteins include MUL1, MARCHV, HUWE1, FBXO7 and RNF185. One mechanism by which USP30 could be suppressing this cascade in a PINK1 dependent manner is by editing ubiquitin from a key trigger protein that serves as the foremost ubiquitylated substrate for PINK1 activity. Note that USP30 like most other DUBs (and E3s) is unable to process phospho-Ub chains, thus the only way its DUB activity can regulate mitophagy initiation is by limiting the PINK1 substrate, ubiquitin itself. We have added a further paragraph expanding on this point in the manuscript on page 5.

The statement that "We did not observe any significant change in the number or distribution of catalase-containing mature peroxisomes, nor in the levels of PEX5, PEX19, Catalase or PMP70 in USP30 depleted or USP30 KO RPE1 cells (Fig. EV3F-I), suggests that the effect on pexophagy is really weak and not reflected by changes in these peroxisomal matrix and membrane proteins.

We have addressed this point above in response to the same comment made at the end of the first paragraph.

Among other related points, the evidence for the integral membrane property is not convincing because both PMPs and catalase (a peroxisomal matrix protein) are largely carbonate insoluble (Fig. 3C).

PMP70 is an integral membrane protein and thus should be resistant to carbonate extraction, as is USP30. Note that carbonate treatment is used to extract peripheral membrane proteins. Comparing our results with those in the literature, we find that in the majority of studies a considerable proportion of catalase remains associated with the pellet under these conditions. However, there clearly is a fraction of catalase that is extracted whilst USP30 (and PMP70) are not. We now also show ACOX1 as a second matrix protein that is more efficiently extracted by sodium carbonate.

This leaves us with a weak phenomenological observation of small effects of mitophagy and pexophagy that comes with no mechanistic insight. No target of USP30 is shown for either mitophagy or pexophagy. The model presented may be an explanation of the data, but that does not mean it is correct.

Referee #4

Ubiquitination was previously established as an important signal that regulates selective autophagy. Marcassa and colleagues report now that USP30, a deubiquitinating enzyme, is involved in regulation of basal mitophagy and pexophagy pathways. USP30 is localized in both organelles and requires its DUB activity for the effect on autophagy pathways. In the case of mitophagy, the novel finding is that USP30 acts upstream of PINK1 and trims Parkin-ubiquitinated substrates during basal mitophagy responses. On the other side, USP30 regulates basal pexophagy independent of PINK1 and Parkin pathways. They also show that the enhancement of both basal mitophagy and pexophagy upon USP30 depletion is dependent on ATG7 indicating that the core autophagy machinery is required for both processes. Importantly they have shown that wild type USP30 but not DUB inactive USP30 can revert the enhanced basal mitophagy and pexophagy upon USP30 knockdown. These are all novel and very important findings in the field of selective autophagy. On a technical side the revised version has offered several new approaches to address most of previously raised valid issues. In cases when there was no better way or approach the authors have provided additional supporting evidence. As such I find this work suitable for publication in EMBO based on both novelty/advance in the field as well as adequate technical quality of assays.

We thank this referee for their positive assessment of our manuscript.

Corresponding Author Name: Sylvie Urbé

Manuscript Number: EMBOR-2017-45595V1